# Peripheral anatomy and central connectivity of proprioceptive sensory neurons in the *Drosophila* wing

**Ellen Lesser[†], Anthony J Moussa, John C Tuthill***

Department of Neurobiology and Biophysics, University of Washington, Seattle, United States

## eLife Assessment

This **important** work describes wing mechanosensory neurons in detail, extending our understanding of sensorimotor processing in the fruit fly. The evidence presented **convincingly** supports the authors' identification of these neurons and leverages state-of-the-art methods to generate a near-complete map of wing mechanosensory circuitry. Overall, this study provides new hypotheses and invaluable tools for investigating proprioceptive motor control of the wing in Drosophila.

**\*For correspondence:**
tuthill@uw.edu

**Present address:** [†]Department of Molecular Genetics and Cell Biology, University of Chicago, Chicago, United States

**Abstract** Recent advances in electron microscopy (EM) and automated image segmentation have produced synaptic wiring diagrams of the *Drosophila* central nervous system. A limitation of existing fly connectome datasets is that most sensory neurons are excised during sample preparation, creating a gap between the central and peripheral nervous systems. Here, we bridge this gap by reconstructing wing sensory axons from the Female Adult Nerve Cord (FANC) EM dataset and mapping them to peripheral sensory structures using genetic tools and light microscopy. We confirm the location and identity of known wing mechanosensory neurons and identify previously uncharacterized axons, including a novel population of putative proprioceptors that make monosynaptic connections onto wing steering motor neurons. We also find that adjacent campaniform sensilla on the wing have distinct axon morphologies and postsynaptic partners, suggesting a high degree of specialization in axon pathfinding and synaptic partner matching. The peripheral location and central projections of wing sensory neurons are stereotyped across flies, allowing this wing proprioceptor atlas and genetic toolkit to guide analysis of other fly connectome datasets.

## Introduction

Fly wings are exquisite, versatile biological actuators. During flight, they sweep back and forth through the air hundreds of times per second to keep the fly aloft. On the ground, flies extend their wings to groom, and males vibrate a wing to attract females during courtship. To accomplish these myriad functions, wing motor control relies on temporally and spatially precise feedback from diverse sensory neurons distributed throughout the wing (**Figure 1A**). Proprioceptive mechanosensory neurons play a particularly important role in flight control (**Pringle, 1957**), as mechanosensory feedback has a shorter latency than visual signals and can, therefore, be used to rapidly adjust wing motion (**Dickerson, 2020**). Wings experience dynamic forces during flight, and proprioceptors encode features of these forces, such as wing bending, twisting, and load (**Dickerson et al., 2021**). *Drosophila* typically beat their wings at 200–250 Hz and can adjust wing kinematics from one stroke to the next (**Dickinson et al., 1993**; **Heide and Götz, 1996**). Thus, muscle contraction must be temporally precise enough to act at these short time scales (**Dickinson and Tu, 1997**). Consistent with the need for rapid feedback,

some motor neurons that control wing steering muscles receive monosynaptic input from wing sensory neurons (*Fayyazuddin and Dickinson, 1999*). However, the peripheral location and identity of the wing sensory neurons that provide feedback to the wing motor system remain largely unknown.

Sensory neurons on the *Drosophila* wing can be grouped into different classes based on their end-organ morphology (*Figure 1B*). The most numerous are the bristles along the wing margin, which include both mechanosensory and chemosensory sensilla (*Hartenstein and Posakony, 1989*; *Palka et al., 1979*). Wing chemosensory neurons can detect external odors and pheromones (*Stocker, 1994*), while mechanosensory bristles can detect the presence of dust particles or mites (*Hampel et al., 2017*; *Li et al., 2016*). Bristles also line the tegula, a cuticular protuberance at the proximal edge of the wing. Apart from the tactile and chemosensory bristles, other sensory neuron classes are presumed to be proprioceptive, in that they monitor the movement and strain of the wing itself. These include campaniform sensilla, chordotonal organs, and hair plates, all of which occur at other locations across the adult fly body, including the legs (*Dinges et al., 2021*; *Field and Matheson, 1998*). Each campaniform sensillum (CS) consists of a single neuron with a dendrite that contacts a cuticular cap, or dome, on the surface of the wing; the CS neuron fires action potentials when the dome deforms (*Chapman et al., 1973*; *Moran et al., 1971*; *Pringle, 1938a*). CS can be found alone or in fields of domes that have similar sizes and orientations (*Cole and Palka, 1982*; *Dinges et al., 2021*). A chordotonal organ (CO) is a cluster of neurons with cap cells that anchor the dendrites to an internal structure, such as a tendon (*Field and Matheson, 1998*). In the wing, they are anchored to inner extensions of the cuticle; for example, the wall of the tegula and the inner wall of the radius (sometimes called the radial vein). A hair plate (HP) is a small, tightly packed cluster of sensory hairs, each of which is innervated by a single mechanosensory neuron (*Pringle, 1938b*). Proprioceptive neurons (CS, CO, and HP) are concentrated proximally, especially along the radius and the tegula (*Figure 1C*). The axons of wing sensory neurons project into the fly's ventral nerve cord (VNC), the invertebrate analog of the spinal cord. Previous work has described the activity of fly leg proprioceptors during walking (*Dallmann et al., 2024*; *Pratt et al., 2026*), but it has been prohibitively challenging to record activity of wing sensory neurons during flight.

Much of what we know about wing sensory neurons comes from developmental studies that used the fly wing as a model to investigate whether axonal morphology is intrinsically determined or extrinsically directed. Some studies used mosaic mutant flies with hindwings in place of halteres to test whether sensory axons would follow haltere-like morphologies or wing-like morphologies once they entered the developing central nervous system (*Ghysen, 1978*; *Palka et al., 1979*). These studies measured morphological similarities between wild-type and mutant axons to uncover their intrinsic developmental programs. Their findings showed that the degree of intrinsic programming was different for single CS and field CS, in that axons from field CS on the mutant hindwings followed similar paths in the VNC to the field CS on wild-type halteres, while the axons of single CS on mutant hindwings retained the morphological characteristics of the wild-type forewing single CS axons (*Palka et al., 1979*). This difference suggests that the field and single CS are endowed with different axon guidance instructions, connect to different postsynaptic partners, and thus may serve distinct functions.

Understanding how central circuits integrate information from wing sensory neurons is key to understanding their function. Connectomics, or dense reconstruction of neurons and synapses from electron microscopy, offers new opportunities for mapping peripheral sensory feedback to the CNS (*Galili et al., 2022*). In this study, we bridge the gap between a VNC connectome and the wing by mapping central axon morphologies to the peripheral structures from which they originate (*Meinertzhagen et al., 2009*). We reconstructed all 490 afferents in the left wing nerve (Anterior Dorsal Mesothoracic nerve, ADMN) in the FANC electron microscopy dataset (*Azevedo et al., 2024*; *Phelps et al., 2021*). Many axon morphologies and their corresponding peripheral end-organs were previously undescribed. We identified genetic driver lines for a subset of these unknown wing sensory neurons and elucidated their peripheral location and anatomy. For example, we identified novel classes of peripheral sensory neurons near the wing hinge and found that CS on the tegula synapse onto the tonic wing b1 motor neuron, suggesting a specialized role in feedback control of flight steering. We also confirmed a long-standing prediction that individual CS from the same field can have distinct axon morphologies (*Palka et al., 1986*). A companion paper that reconstructed haltere CS axons in the connectome identified a similar organization (*Dhawan et al., 2026*). Overall, knowing the relationships between peripheral neuroanatomy, axon morphology, and downstream connectivity

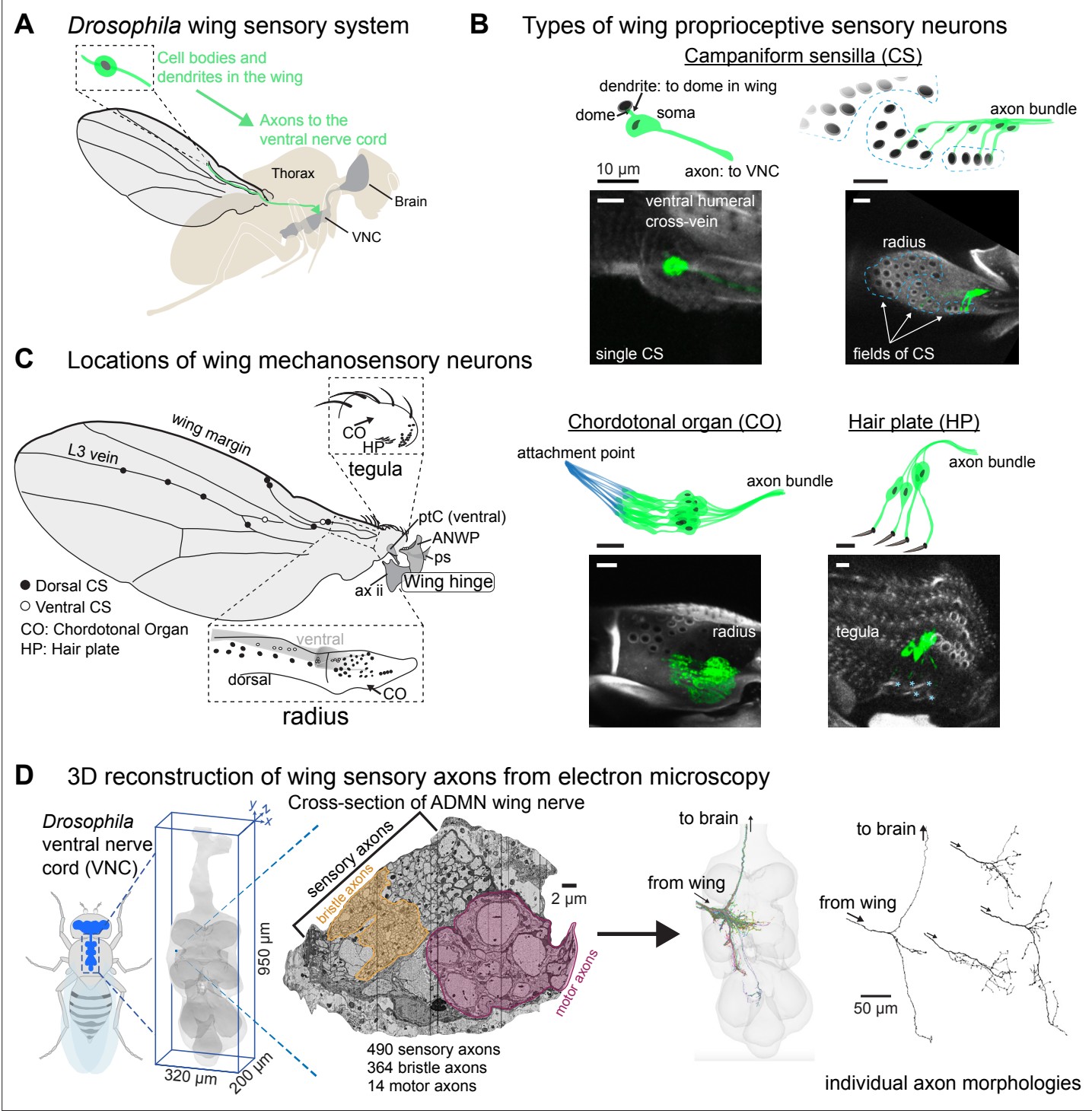

**Figure 1.** Proprioceptive neurons on the *Drosophila* wing. (**A**) The cell bodies and dendrites of sensory neurons are in the periphery, on the wing and wing hinge, and their axons project to the ventral nerve cord (VNC). Before entering the VNC, the sensory axons fasciculate together and enter through the Anterior Dorsal Mesothoracic Nerve (ADMN). (**B**) Proprioceptors on the wing include campaniform sensilla (CS), chordotonal organs (CO), and a hair plate (HP). Each campaniform sensillum dome is innervated by a single sensory neuron, as is each hair in a HP. A CO is made up of a group of sensory neurons with supporting cells that fix them to the underside of the cuticle (shown in blue). Blue asterisks (bottom right) indicate a single external hair in the HP. Images show the membrane-bound fluorescent label mCD8::GFP to highlight each proprioceptor type. See *Appendix 1—table 3* for details on which proprioceptors are labeled by which driver lines, driver lines for representative images in this panel are: single CS (12C07-GAL4); field CS (10G03-GAL4); CO (15F10-GAL4); HP (16C09-GAL4). Scale bars are 10 μm. (**C**) Location of sensory neurons on the wing and wing hinge. The location of sensory neurons and the number of CS in each field are based on confocal images and a prior study (*Dinges et al., 2021*). A subset of sclerites and

*Figure 1 continued on next page*

*Figure 1 continued*

other structures that make up the wing hinge are included as landmarks: pterale C (ptC), the anterior nodal wing process (ANWP, which also features three CS), the parascutal shelf (ps), and the second axillary (ax ii). (**D**) We reconstructed each sensory axon in the ADMN wing nerve to visualize its full morphology and analyze downstream connectivity in the VNC. More information on each of these steps is in *Azevedo et al., 2024*. In the nerve cross-section, the motor domain and margin bristle domains are highlighted by outlined yellow and mauve masks.

The online version of this article includes the following figure supplement(s) for figure 1:

**Figure supplement 1.** Pipeline for matching 3D reconstructed axons to sensory neurons on the wing and wing hinge.

to wing motor neurons provides a foundation for investigating proprioceptive sensing and motor control of the fly wing.

## Results

### Comprehensive reconstruction of wing axons in the FANC connectome

We reconstructed all axons in the left ADMN using an EM dataset of the VNC of a female adult fly (FANC; *Figure 1D*; also see **Methods**; *Azevedo et al., 2024*; *Phelps et al., 2021*). For each automatically segmented neuron, we used the software interface Neuroglancer to manually proofread the major branches, as well as all branches that could be reliably attached (**Methods**). In the left ADMN, we identified 490 sensory axons and 14 motor axons. Axons were identified as sensory if they did not attach to a cell body in the VNC (**Methods**). The total number of axons is slightly higher than previously reported counts from cross sections of the wing nerve (455–465 axons, *Edwards et al., 1978*). Of these afferents, we classified 364 as wing margin bristle axons based on their ventral projections (*Palka et al., 1979*). Of the 126 non-bristle afferents, we identified 64 axon morphologies from published images of dye-fills (*Appendix 1—table 1*; *Burt and Palka, 1982*; *Ghysen, 1980*; *Ghysen, 1978*; *Kays et al., 2014*; *Koh et al., 2014*; *Lu et al., 2012*; *Palka et al., 1986*; *Palka et al., 1979*; *Thistle et al., 2012*; *Whitlock and Palka, 1995*). Of the 62 remaining axons previously unidentified in the literature, we identified sparse GAL4 lines in the FlyLight collection (*Jenett et al., 2012*) that labeled axon morphologies that resembled the reconstructed axons from FANC, crossed these lines to a fluorescent reporter, and then imaged the wing and wing hinge to visualize expression (*Figure 1—figure supplement 1*). Using this strategy, we successfully identified 50 of the 62 previously unidentified morphologies.

We reconstructed postsynaptic partners of sensory neurons until at least 70% of the output synapses from each sensory neuron were attached to proofread neurons (*Figure 2A*). Sensory axons make direct synapses onto motor neurons, other sensory neurons, VNC intrinsic neurons, and interneurons that ascend to the brain. To identify clusters of sensory axons with similar postsynaptic connectivity, we used a pairwise measure of cosine similarity, where a score of 1.0 indicates that the two neurons contact the same partners with the same proportion of synapses. We then ordered the neurons via agglomerative clustering, which revealed clusters of neurons with similar morphologies (*Figure 2B*). The cosine similarity of axon pairs within each cluster was significantly higher than across clusters (*Figure 2B*, **inset** (permutation test; 10,000 permutations, observed difference = 0.34, $p<0.05$)). *Figure 2C–E* shows the axon morphology of each cluster, organized by peripheral class. In the remainder of the paper, we focus on identifying the novel sensory neuron classes in *Figure 2C*.

Proximal CS axons are characterized by three branches: one short branch projects to the tectulum and two long branches project anteriorly to the brain and posteriorly to the haltere neuropil (*Ghysen, 1980*). There are ~36 proximal CS on the wing, and we found 38 axons in the EM dataset that followed this pattern. Previous dye fills of distal CS revealed axons that do not ascend to the brain and instead send two processes to the posterior VNC (*Ghysen, 1980*). There are ~17 distal CS on the wing, and we found 15 axons that match this pattern. We also identified five ascending axons that resemble the small CS morphology, although they are missing a posterior branch. Overall, our comprehensive reconstruction revealed many morphological subgroups with overlapping postsynaptic partners, suggesting a high degree of integration within wing sensorimotor circuits.

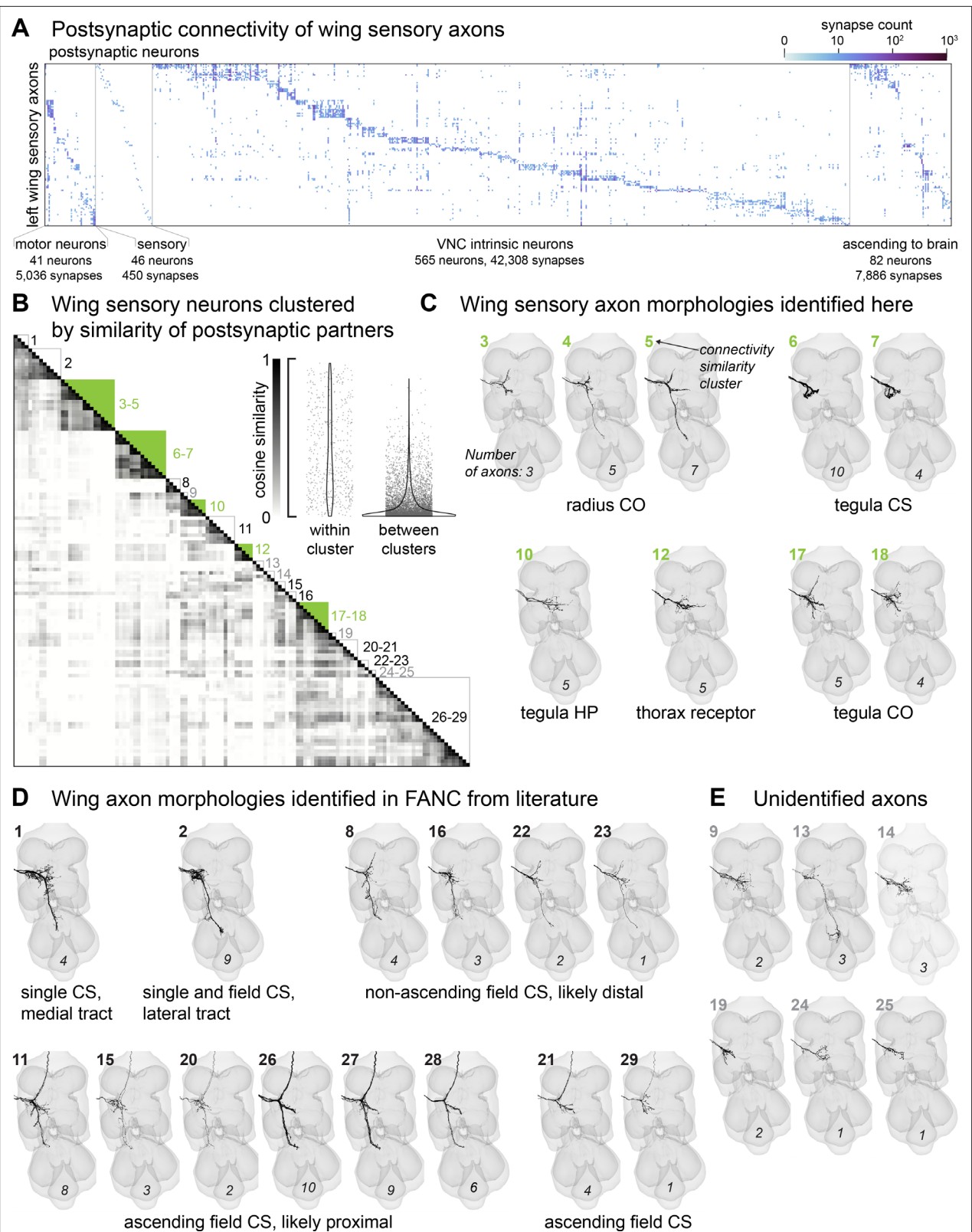

**Figure 2.** Postsynaptic connectivity and morphology of wing sensory axons. (**A**) Connectivity matrix based on the left wing proprioceptors and postsynaptic neurons in the ventral nerve cord (VNC). Only partners with at least five synapses from a single proprioceptor are shown. For visualization simplicity, we do not show: (1) a descending neuron that is postsynaptic to sensory neurons (0.1% of the proprioceptive outputs), (2) a single non-motor efferent neuron (0.1%), and (3) unproofread or fragment neurons (9.7%). Postsynaptic neurons are classified as either motor neurons, sensory neurons,

*Figure 2 continued on next page*

*Figure 2 continued*

VNC intrinsic neurons, or ascending neurons (axons project to the brain). Within each class, postsynaptic neurons were then sorted according to which wing proprioceptor they receive the most synapses from. The number of synapses is displayed on a log scale. (**B**) Cosine similarity matrix of the 126 left wing axons not from margin bristles. Axons are ordered by agglomerative clustering. The inset shows pairwise similarity scores for each pair of axons. Within-cluster similarity is greater than between clusters (permutation test; 10,000 permutations, observed difference = 0.34, *p*<0.05). Boxes in the matrix indicate clusters of axons with similar morphology, with the number next to each cluster indicating the morphology clusters in (**C–E**). Filled green boxes indicate morphologies identified in this study. See *Appendix 1—table 1* and **Methods** for details on matching axon morphologies to prior literature.

## Connectivity reveals a potential role for the tegula in flight control

Due to the need for rapid sensory feedback necessary for flight control, we were especially interested in identifying axons with monosynaptic connections onto wing motor neurons. We found that 34 of 62 previously uncharacterized axons synapse onto wing steering motor neurons (*Figure 3A–B*). Of these, one group of axons synapses directly onto the well-characterized b1 motor neuron, which innervates the b1 muscle to help stabilize pitch during flight (*Whitehead et al., 2022*). The b1 motor neuron fires on nearly every wing stroke, and input from wing afferents sets the phase of its activation (*Fayyazuddin and Dickinson, 1999*; *Heide and Götz, 1996*). Notably, the input to the b1 motor neuron from ipsilateral wing and haltere axons is clustered around the putative spike initiation zone (*Figure 3B*), as has previously been reported based on axonal spatial overlap (*Chan and Dickinson, 1996*). This synaptic organization may be a structural mechanism for facilitating rapid modulation of b1 activity based on sensory feedback.

The wing sensory axons that synapse onto the b1 motor neuron have not been previously characterized (*Figure 3C*). They terminate shortly after entering the VNC and do not branch more extensively. We observed an unexpected ultrastructural feature in these axons: their terminals contain very densely packed mitochondria compared to other cells (*Figure 3D*). This feature is also present in interneurons that make electrical connections (*Trimarchi and Murphey, 1997*) to the b1 motor neuron (*Figure 3E*). We speculate on this ultrastructure further in the **Discussion**.

To identify the peripheral identity of these axons (*Figure 3F*), we found a driver line that labeled this population (*Figure 3G*; *Jenett et al., 2012*) and crossed it to a fluorescent reporter. Imaging the wing revealed that the population of short axons that directly synapse onto a subset of wing steering motor neurons originates from a field of CS on the tegula (*Figure 3H*). This finding suggests that the tegula may play a previously underappreciated role in flight control, particularly in regulating the tonically firing muscle b1 and a tonically firing muscle from another motor module, i2 (*Figure 3A*).

## Hair plate (HP) on the tegula

A group of five axons branch dorsally and ventrally as they enter the VNC and cross the midline (*Figure 4A*). We identified a sparse driver line that labeled these neurons (*Figure 4B*) and found that their corresponding cell bodies were in the tegula. There is a row of short stubby hairs on the dorsal face of the tegula (*Figure 4C*), resembling the HPs found at leg joints that are activated at extreme joint positions (*Pratt et al., 2026*; *Pringle, 1938b*; *Trimarchi et al., 1999*). The role of this tegula HP in wing sensation is unknown, although their peripheral morphology had been previously described (*Fudalewicz-Niemczyk, 1963*).

## Chordotonal organ in the tegula

Two groups of axons with similar postsynaptic partners branch broadly throughout the tectulum without crossing the midline (*Figure 5A*). Using a sparse driver line (*Figure 5B*), we found that the cell bodies belong to an internal structure within the tegula (*Figure 5C*). We counted ~14 neurons in this structure, which separate into two bundles that attach to different points on the distal, anterior end of the tegula. Neurons in the chordotonal organ (CO) in the tegula are not labeled by *iav*-GAL4, unlike many other COs elsewhere in the body (*Figure 5D*; *Kwon et al., 2010*). They do, however, have actin-rich cap cells that are characteristic of other COs and not present in other mechanosensory neurons such as CS or HPs (*Figure 5E*; *Field and Matheson, 1998*). A subset of tegula CO neurons are labeled by NompC-GAL4, suggesting that they do express mechanosensory channels other than *iav* (*Figure 5F*). Because we only identified nine axons, vs. 14 cell bodies, some tegula CO axons might have a different morphology.

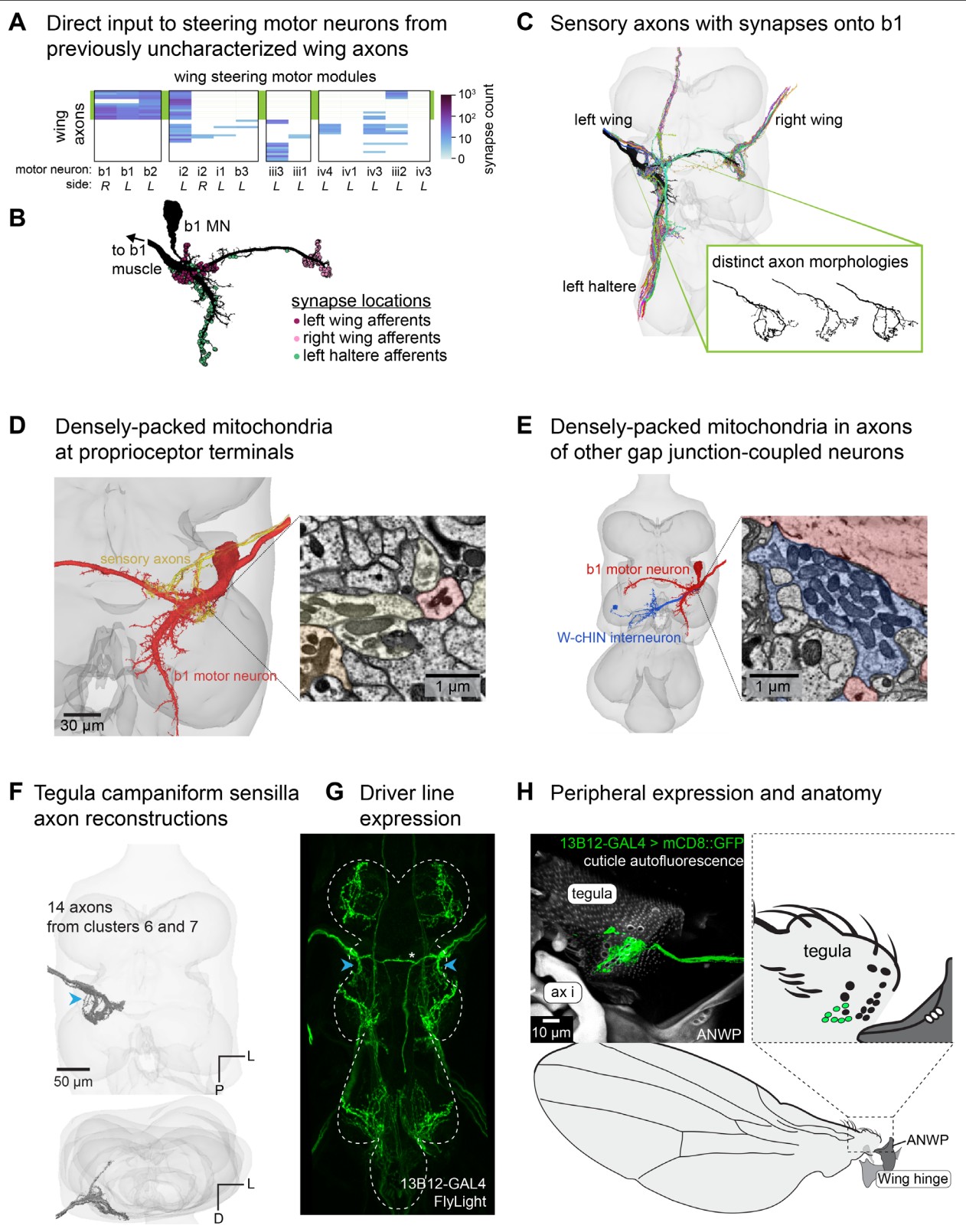

**Figure 3.** Campaniform sensilla (CS) on the tegula target the tonic wing steering motor neuron b1. (**A**) Connectivity between previously uncharacterized wing sensory axons and wing steering motor neurons. Wing steering motor neurons (columns) are grouped by motor modules, which are groups of motor neurons that receive a high degree of synaptic input from shared presynaptic partners and are therefore likely to be co-activated (*Lesser et al., 2024*). The green box behind the plot highlights a group of axons with a shared morphology, discussed in the rest of the figure. (**B**) The left b1

*Figure 3 continued*

motor neuron with circles showing predicted synapse locations from the Female Adult Nerve Cord (FANC) electron microscopy (EM) volume. (**C**) 3D reconstructions of the left b1 motor neuron (black) and all the sensory axons from which it receives direct synaptic input. Inset: three example individual axons from the left wing to demonstrate the variation in axon branching. (**D**) Ultrastructure of putative electrical synapses: these sensory axons feature densely packed mitochondria at terminals near the b1 motor neuron. (**E**) A similarly high density of mitochondria is also seen at axon terminals of a wing contralateral haltere interneuron (w-ChiN), which likely have electrical synapses onto b1 based on dye-fill experiments (*Trimarchi and Murphey, 1997*). (**F**) Axon branching pattern in VNC. Axons are from two morphological clusters (#6 and #7 from *Figure 2*). Below: rotated view of the VNC. (**G**) Maximum projection from FlyLight Z-stack of images of the driver line 13B12-GAL4. The projection crossing the midline (indicated by a white asterisk) is from a different sensory neuron that enters through the Posterior Dorsal Mesothoracic Nerve and innervates a thorax bristle. (**H**) Expression in the periphery. Maximum projection from confocal Z-stack showing sensory neurons that innervate the CS field on the tegula. The driver line also labels two tegula HP hairs, but their axon morphology is distinct (see *Figure 4*). Wing hinge abbreviations: anterior nodal wing process (ANWP), first axillary (ax i).

## Chordotonal organ in the radius

Axons with three distinct morphologies share a characteristic branch that passes laterally through the wing neuropil (*Figure 6A*). One group of axons extends a long process into the haltere neuropil, and another crosses the midline. By imaging sparse driver lines, we found that these axons come from neurons that make up a CO in the radius (*Figure 6B–C*). These neurons are distinguishable from the CS neurons in the radius because they do not have a dendrite that reaches toward the surface of the vein to innervate a dome. Instead, the cell bodies sit on the posterior side of the radius, and their dendrites and cap cells insert on the anterior side of the radius. These neurons all attach to the same point on the wing vein (*Figure 6D*), so they are likely subject to the same mechanical forces, allowing the CO to send parallel information to multiple regions of the VNC.

## Thorax sensor near the wing hinge

Five axons each extend a single process through the dorsal tectulum, and two of the axons cross the midline (*Figure 7A*). These axons originate from a cluster of five neurons in the thorax beneath the

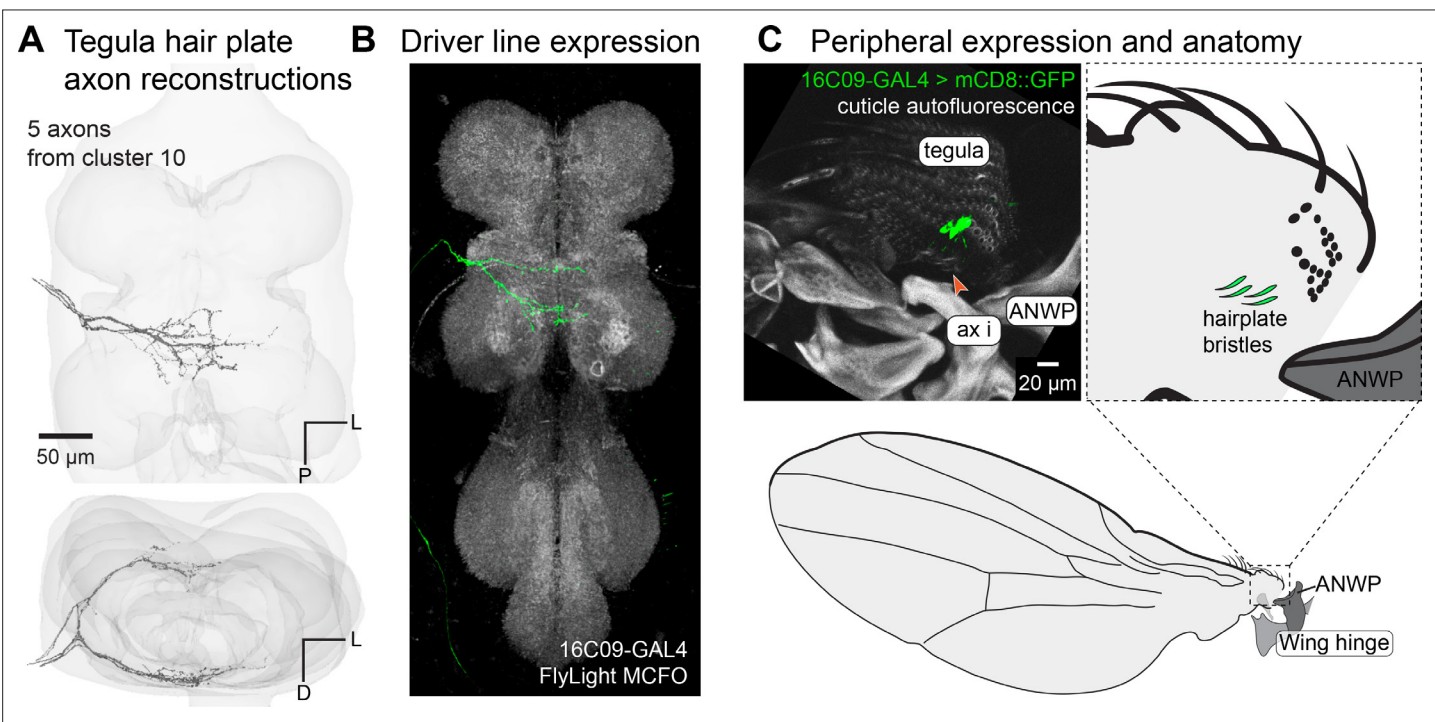

**Figure 4.** Tegula hair plate. (**A**) 3D reconstructed axons. Above: population of axons with similar morphology (black) and ventral nerve cord (VNC) volume (gray). Below: rotated view to show how the axons split to scoop around the dorsal and ventral edges of the wing neuropil. (**B**) Axon branching pattern in VNC. Axons are from morphological cluster #10 in *Figure 2*. Maximum projection from the FlyLight MCFO collection of the driver line 16C09-GAL4. (**C**) Expression in the periphery. Maximum projection from confocal Z-stack showing sensory neurons that innervate the hairs of the tegula hair plate. Red arrow indicates an external hair plate hair. Wing hinge abbreviations: anterior nodal wing process (ANWP), first axillary (ax i).

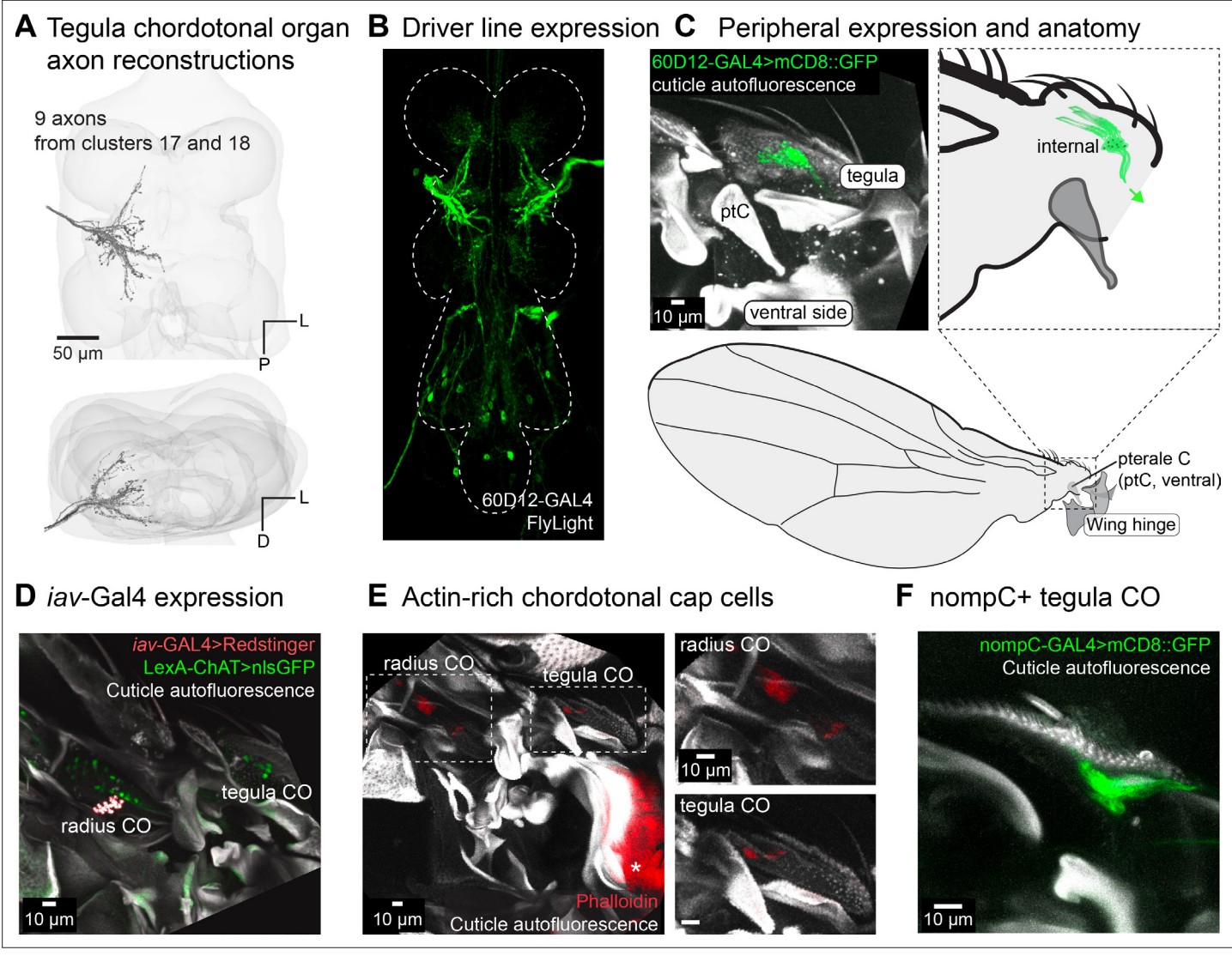

**Figure 5.** Tegula chordotonal organ. (**A**) 3D reconstructed axons. Axons are from two morphological clusters (#17 and #18 in **Figure 2**). (**B**) Axon branching pattern in ventral nerve cord (VNC). Maximum projection from FlyLight Z-stack of images of the driver line 60D12-GAL4. (**C**) Expression in the periphery. Maximum projection from confocal Z-stack showing sensory neurons that innervate the chordotonal organ in the tegula. There are two clusters of neurons, which are differentiated by their separate attachment points within the tegula. 60D12-GAL4 labels neurons from both clusters. (**D**) Maximum z-projection of the proximal wing co-labeling *iav*-GAL4 with ChAT-LexA. ChAT-LexA labels nearly all sensory neurons (green, nuclear stain) and *iav*-GAL4 labels the radius chordotonal organs (CO) but not the tegula CO (red, nuclear stain). (**E**) Phalloidin labels the actin-rich cap cells that are part of chordotonal organs. Asterisk indicates muscle that is also labeled by phalloidin. (**F**) nompC-GAL4 labels all sensory neurons in the tegula, including the chordotonal organ.

wing hinge near the parascutal shelf, just medial to the anterior nodal wing process (**Figure 7B–C**). Other than the three CS on the anterior nodal wing process, these are the only cells near the wing hinge labeled by the *ChAT*-GAL4 driver line, which targets nearly all peripheral sensory neurons (**Figure 7C**; **Yasuyama and Salvaterra, 1999**). As with all the anatomically defined populations of axons, the function of these novel wing hinge sensory neurons will require physiological measurements, but based on their location, they may signal wing opening and closing. We found no evidence for sensory neurons innervating pterale C (**Figure 7D**), a wing hinge sclerite that was previously thought to contain sensory receptors (**Miyan and Ewing, 1984**), although axons from the radius travel directly beneath pterale C.

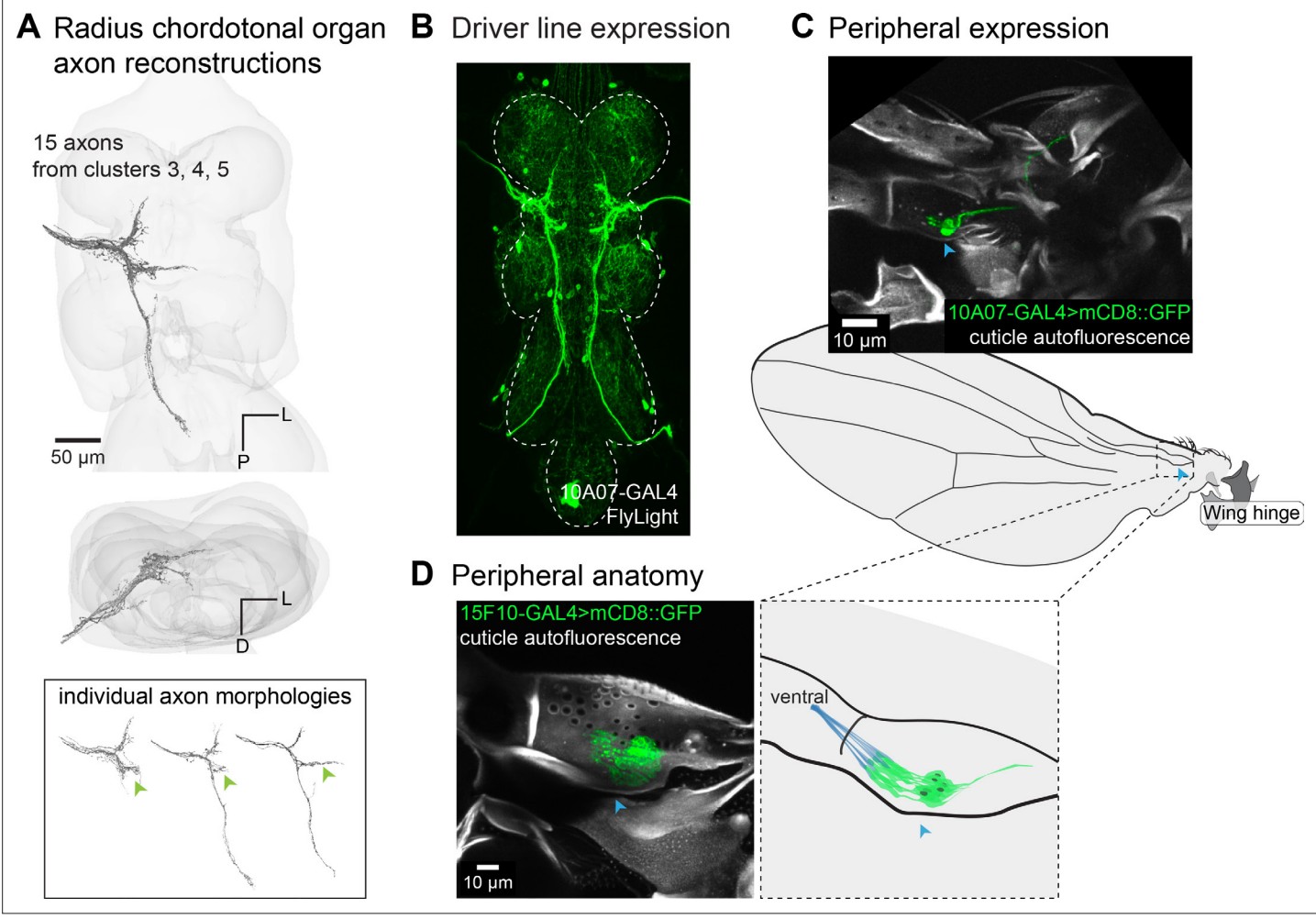

**Figure 6.** Radius chordotonal organ. (**A**) 3D reconstructed axons. Axons are from the morphological clusters #3, #4, and #5 in *Figure 2*. Green arrow indicates the characteristic lateral projection found in each neuron. (**B**) A sparse driver line, 10A07-GAL4, labels a subset of neurons that make up the radius chordotonal organ. For other driver lines that label radius chordotonal neurons, see *Appendix 1—table 3*. (**C**) Peripheral expression of 10A07-GAL4>UAS-mCD8::GFP. (**D**) Peripheral anatomy of the radius chordotonal organ, which is better shown by a broad driver line, 15F10-GAL4>UAS-mCD8::GFP. The radius chordotonal organ attaches to the ventral inner wall of the radius by cap cells (blue). A blue arrow is shown across the confocal images and cartoons to orient to the 'pocket' in the radius near the chordotonal organs (CO) cell bodies.

## The axons of adjacent campaniform sensilla (CS) are morphologically distinct

Previous work uncovered morphological diversity across CS axons — subsets of CS axons follow different tracts and some ascend to the brain while others do not (*Palka et al., 1979*). It was unclear, however, whether there is morphological diversity of axons that originate from the same CS field. A field of CS is defined as having the same size, circularity, orientation, and general location on the wing (*Cole and Palka, 1982*). Single CS can be either rapidly or slowly adapting (*Dickinson and Palka, 1987*), but whether all CS within a field have the same firing properties remains unknown. It is also not known whether all CS within a field connect to the same postsynaptic partners. Answering this question could provide insight into the function of spatially clustered CS, for example, whether they underlie a population code or transmit signals in parallel to distinct downstream circuits.

To determine the morphological similarity of axons that innervate CS in the same field, we identified GAL4 driver lines that sparsely expressed in the wing nerve (less than five ADMN axons) and imaged their expression in the wing. We found many lines that label subsets of CS across multiple fields (*Appendix 1—table 3*) and fortuitously found three driver lines that label one to two separate CS in the ventral radius C field (v.Rad.C, *Figure 8A*). For these three lines, we compared VNC expression of

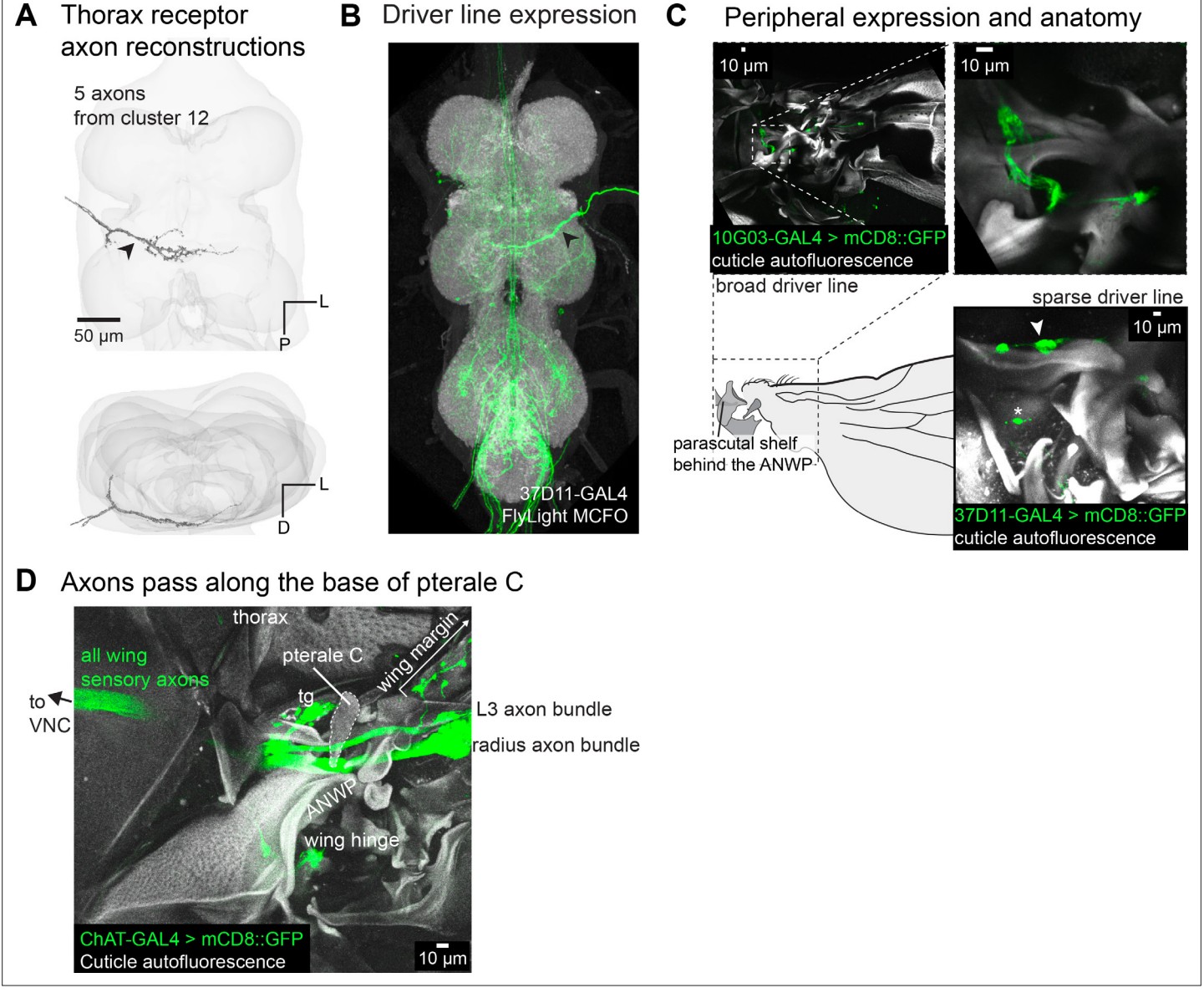

**Figure 7.** Sensory axons near the wing hinge. (**A**) 3D reconstructed axons. Axons belong to the morphological cluster #12 from **Figure 2**. (**B**) Axon branching pattern in ventral nerve cord (VNC). Maximum projection from the FlyLight MCFO collection of the driver line 37D11-GAL4. (**C**) Expression in the periphery. Top: maximum projection from confocal Z-stack of a broader driver line, 10G03-GAL4, to show the morphology of the sensory neurons at the base of the parascutal shelf. Below: maximum projection from confocal Z-stack of the sparse driver line 37D11-GAL4>UAS-mCD8::GFP showing neurons labeled at the base of the parascutal shelf. The asterisk marks an innervated bristle on the thorax. (**D**) Pterale C is not an innervated sclerite. Pterale C was previously predicted to be innervated based on experiments in which an electrode placed at the base of pterale C recorded signals in response to wing vibration (**Miyan and Ewing, 1984**). We found no neurons innervating pterale C, but we did observe that the axon bundle from the radius passes directly under pterale C, which could explain previously published results.

axons from the ADMN using the FlyLight Multi-Color Flp-Out (MCFO) collection (**Figure 8B–D**; **Meissner et al., 2023**). In a driver line that expresses in two v.Rad.C neurons (CS 2 and 4), the VNC contains two distinct axon morphologies originating from the ADMN nerve (**Figure 8D**, **row 1**). Both axons possess a process that ascends to the brain, but one also projects down to the haltere neuropil. In a second driver line that also labels the second CS in v.Rad.C, we observed the same axon morphology that ascends to the brain but does not reach the haltere neuropil (**Figure 8D**, **row 2**). A third driver line that expresses in the third CS in v.Rad.C contains a non-ascending wing axon with two posterior projections (**Figure 8D**, **row 3**). These results show that neurons that innervate adjacent campaniform sensilla within the same field can have different axon morphologies. Each of the three v.Rad.C

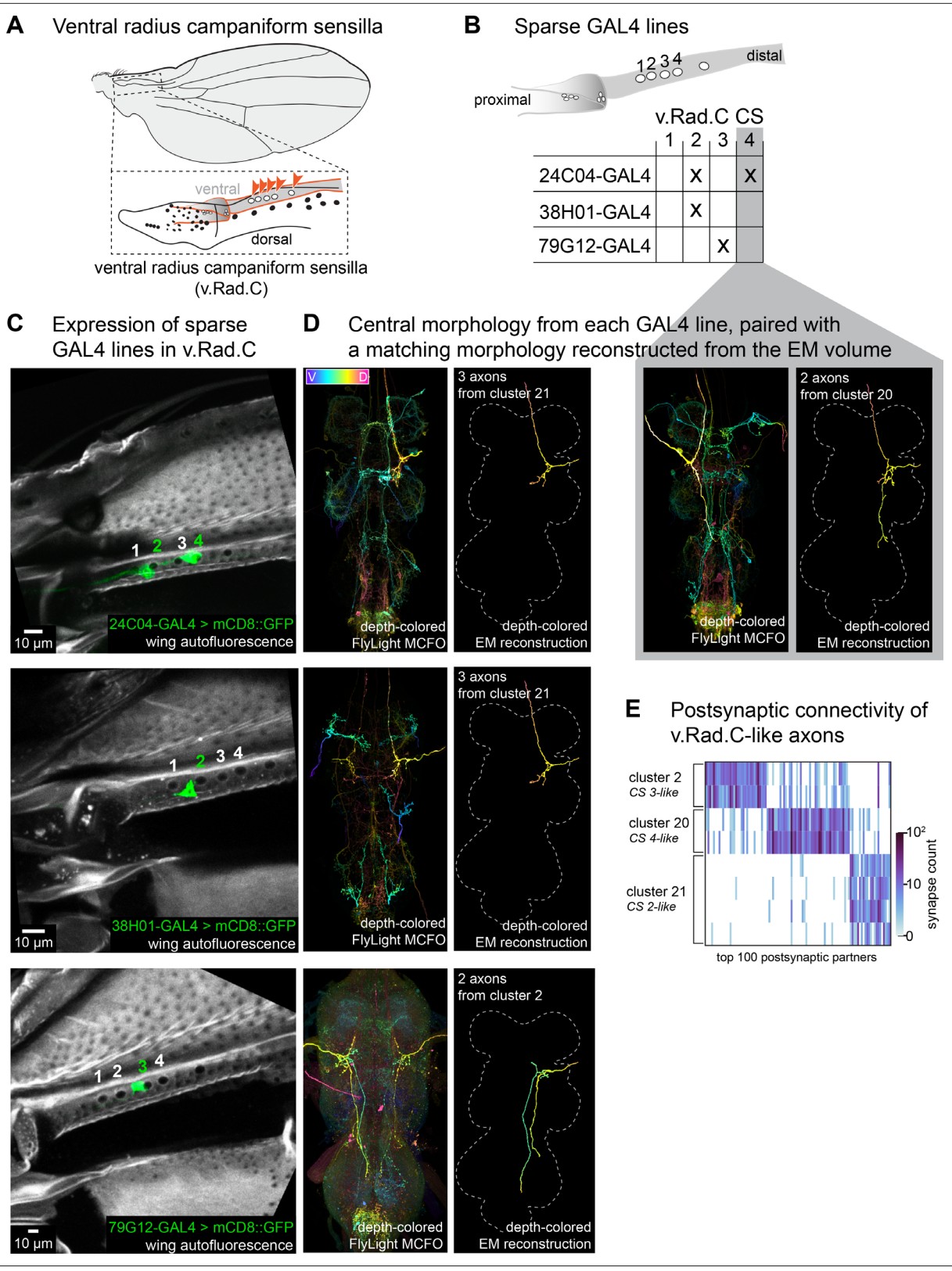

**Figure 8.** Campaniform sensilla in the same field have unique axons. (**A**) The ventral radius C (v.Rad.C) field of campaniform sensilla is on the ventral side of the more distal part of the radius. The field has four to five domes, the fifth dome is proposed to be its own individual dome as it is farther apart from the other four and its orientation is slightly different (***Dinges et al., 2021***). (**B**) Summary of the campaniform sensilla (CS) within v.Rad.C that are labeled by sparse GAL4 lines shown in **C**. (**C**) Peripheral expression in specific campaniform sensilla from sparse driver lines (each row). Maximum

*Figure 8 continued on next page*

*Figure 8 continued*

projection from confocal Z-stack showing expression in the periphery from each sparse driver line. CS in v.Rad.C are labeled 1-4 as in (**B**) to show which CS is innervated in each image. (**D**) Pairs of images showing (left) A depth-colored single channel MCFO Z-stack from the FlyLight collection (*Meissner et al., 2023*), with the wing axon highlighted in the image. The contrast of z-sections was optimized to emphasize visual clarity of wing axons; see **Methods** for details. (Right) The reconstructed axon from electron microscopy (EM) that best matches the morphology, depth-colored and aligned to the same template as the FlyLight images. (**E**) Postsynaptic connectivity of axons with morphologies that match those found for v.Rad.C. Postsynaptic connectivity is more similar for axons with similar morphologies than from the same CS field.

axons falls into a cluster of morphologically similar axons that connect to similar postsynaptic neurons (*Figure 8E*, clusters 2, 20, and 21, respectively, from *Figure 2*). The additional axons in each cluster likely originate from CS in other fields, on different parts of the wing. Overall, our results suggest that adjacent CS neurons in the same field connect to different target neurons in the VNC, and that the CS from different fields can connect to common postsynaptic targets.

## Discussion

In this project, we reconstructed each sensory axon in the left ADMN wing nerve, as well as their post-synaptic partners, from an EM dataset of the *Drosophila* VNC (*Azevedo et al., 2024*; *Phelps et al., 2021*). We matched previously unidentified axon morphologies to corresponding sensory structures on the wing by imaging the expression of genetic driver lines. To make this information accessible to the community, we provide a library of confocal Z-stacks and an annotation table linked to the FANC connectome dataset (see **Methods**). We also include a reference (*Appendix 1—table 1*) to match peripheral sensory structures identified here to neuron nomenclature established in the Male Adult Nerve Cord (MANC) connectome, in which wing sensory axons are proofread but few are annotated by peripheral identity (*Marin et al., 2024*). The relative ease with which we were able to match sensory neurons across the two datasets based on morphology suggests that there were no obvious sexual dimorphisms in these neurons. A closer examination across more than two datasets is needed to rigorously test this observation.

### Integrating connectomics with previous research

Sensory neurons in the *Drosophila* wing have been a useful model system for investigating the relative contributions of intrinsic regulation vs. extrinsic signaling in the determination of axon morphology (*Ghysen, 1980*; *Palka et al., 1979*). Past work identified the axonal morphologies of wing CS and differences in the genetic programs guiding axon development for CS in fields and single CS (*Ghysen, 1980*; *Palka et al., 1979*). Here, we build on this knowledge using cell-type-specific genetic tools and connectomics to create a more complete map of the wing sensory apparatus and central sensorimotor circuits. By reconstructing axons from an electron microscopy volume, we found novel wing sensory axon morphologies, many of which originated from internal sensory structures that were inaccessible with previous dye-fill techniques, such as the previously uncharacterized CO axons.

Most wing CSs are organized into fields, in which domes of roughly the same size and shape cluster together. By finding sparse driver lines that label only one or two CS in the ventral radius C field (v.Rad.C), we found that CS in the same field can have distinct axon morphologies. This finding is consistent with our unsuccessful attempts to build split-GAL4 driver lines that specifically label CS from a single field by intersecting lines that label the same axon morphology. This organization may offer a functional advantage by sending similar signals to multiple regions of the CNS in parallel; viewed from a different perspective, postsynaptic neurons may rapidly integrate information across different CS fields. One possibility is that CS axon morphology is more closely linked to neuronal intrinsic properties (e.g., slowly vs. rapidly adapting). We hypothesize that the morphological clusters we identified (*Figure 8D–E*) are similarly tuned CS neurons distributed throughout different fields across the wing. A concurrent study focused on the central and peripheral organization of haltere mechanosensory neurons found that common central axon morphologies map across multiple CS fields on the haltere (*Dhawan et al., 2026*), consistent with our prediction for the wing.

One advantage of the dye-fill technique is that it is sometimes feasible to repeatedly label the same exact neuron across individuals. By comparing axon morphologies across individuals, researchers have found differences in small branches that extend from the primary neurite of the neuron (*Kays et al.,*

2014; *Palka et al., 1986*). As more connectomic datasets become available, we will be able to ascertain if this morphological variation is also reflected by variation in downstream connectivity. Understanding relationships between morphology and connectivity across individuals, and even across species, will provide a framework for deciphering the developmental logic that governs the formation of sensorimotor circuits.

## Insights into motor control based on connectivity and structural anatomy

### Direct sensory input to wing motor neurons

Many wing sensory neurons, such as those originating from the tegula and near the wing hinge, synapse onto wing motor neurons, providing a mechanism for rapid feedback-driven motor control. Four steering wing motor neurons receive over 10% of their synaptic input from wing and haltere sensory neurons (*Lesser et al., 2024*). Each of these four motor neurons also fires tonically during flight, whereas other steering motor neurons burst during maneuvers, such as turning (*Lindsay et al., 2017*). Integrating direct sensory input provides a mechanism for low latency motor control, which is important given the short timescales required for Dipteran flight (*Dickerson, 2020*). We found that the CS on the tegula provided the most direct feedback onto wing steering motor neurons (*Figure 3*). Future work is needed to understand what these sensory neurons encode. Based on their direct connections and thick axons, we speculate that they set the firing phase of tonically active muscles, such as b1 (*Fayyazuddin and Dickinson, 1999*; *Heide, 1983*).

### The tegula as a major sensory structure

Although the tegula has been studied for its role in locust flight control, it has largely been neglected in Dipteran literature. In both locusts and flies, the tegula features CS, a CO, and mechanosensory hairs (*Fudalewicz-Niemczyk, 1963*; *Wolf, 1993*). Feedback from the locust tegula resets the phase of wing elevation and the forewing tegulae are only necessary to maintain the flight rhythm if the hindwing tegulae are compromised (*Büschges et al., 1992*). Unlike in locusts, Dipteran flight muscles are asynchronous: wing elevation and depression are driven biomechanically rather than by neural activation of individual strokes (*Deora et al., 2015*). Despite the different feedback demands between asynchronous and synchronous flight, all flying insects share an evolutionary history (*Gau et al., 2023*) and wings experience similar forces, which are optimally sensed from particular locations (*Weber et al., 2021*). Examining how the tegula contributes to flight control across taxa offers opportunities to better understand the evolutionary pressures shaping mechanosensory feedback in flying insects.

### Mechanosensation at the wing hinge

In locusts, feedback from stretch receptors embedded in the wing hinge can directly modify wingbeat frequency (*Gettrup, 1962*; *Wilson and Gettrup, 1963*). In *Diptera*, however, there is scant literature on a sensory organ embedded near the wing hinge (*Hertweck, 1931*). The only putative sensory structure at the wing hinge is the sclerite pterale C (*Miyan and Ewing, 1984*). This hypothesis was based on spikes recorded from a sharp electrode placed at the base of pterale C in response to wing vibration. We found no cells labeled by a pan-sensory neuron driver (*ChAT*-GAL4) at the base of, or innervating, sclerite pterale C. We did, however, observe that the entire nerve of sensory axons from the radius passes through the base of pterale C (*Figure 7D*), and thus speculate that action potentials traveling along this nerve are likely what was being recorded in that prior study. We also observed a cluster of previously unreported cells labeled by *ChAT*-GAL4 near the parascutal shelf, which was also labeled by several sparse driver lines (see *Appendix 1—table 3*, column 'thorax receptor'), and may be the same structure described previously (*Hertweck, 1931*).

### A potential metabolic specialization for flight circuitry

In addition to morphology and connectivity, the EM volumes of the VNC reveal the ultrastructure of neurons and synapses. While reconstructing neurons in FANC, we noticed an unusual density of mitochondria in the axon terminals of specific wing sensory and premotor neurons (*Figure 3D–E*). We did not notice equivalent specializations in prior projects that reconstructed and analyzed leg proprioceptors (*Lee et al., 2025*) and premotor neurons (*Lesser et al., 2024*). Notably, some of

the terminals with dense mitochondria were at sites of known gap-junction coupling (*Trimarchi and Murphey, 1997*). In the fly VNC, electrical synapses are often accompanied by chemical synapses, which may cooperate to ensure low latency signal transmission (*Fayyazuddin and Dickinson, 1996*). In the adult fly brain, however, sites of gap junction coupling, such as the lobula plate tangential cells (*Ammer et al., 2022*), do not exhibit particularly high concentrations of mitochondria (*Sager et al., 2025*). Therefore, we speculate that the density of mitochondria we observe may be a specialization to support the low-latency feedback necessary for controlling flight. More work is needed to understand the significance and function of high mitochondrial density in wing sensorimotor circuits.

### Remaining gaps

Although it is the most comprehensive to date, our atlas of wing sensory neurons is not complete. There were six axon morphologies (12 total axons) from the connectome that we could not reliably map to peripheral structures. There were also several peripheral structures whose axon morphologies we could not identify, such as the three CS on the anterior nodal wing process and the two large CS on the tegula. Furthermore, some of the uncharacterized axon morphologies likely belong to the tegula and radius COs, both of which had more cell bodies than identified axons. These gaps highlight the need for complementary approaches, such as combining small-scale experimental approaches with large-scale comprehensive datasets, to fully characterize the wing's sensory landscape.

## Methods

**Key resources table**

| Reagent type (species) or resource | Designation | Source or reference | Identifiers | Additional information |
|---|---|---|---|---|
| Antibody | Alexa Fluor Phalloidin 647 | Thermo Fisher | Thermo Fisher A22287 | 1:50 in PBST |
| Genetic reagent (*D. melanogaster*) | 10A07-GAL4 | Bloomington *Drosophila* Stock Center | RRID:BDSC_48435 | w[1118]; P{y[+t7.7] w[+mC]=GMR10A07-GAL4}attP2 |
| Genetic reagent (*D. melanogaster*) | 10F07-GAL4 | Bloomington *Drosophila* Stock Center | RRID:BDSC_48266 | w[1118]; P{y[+t7.7] w[+mC]=GMR10F07-GAL4}attP2 |
| Genetic reagent (*D. melanogaster*) | 10G03-GAL4 | Bloomington *Drosophila* Stock Center | RRID:BDSC_48271 | w[1118]; P{y[+t7.7] w[+mC]=GMR10G03-GAL4}attP2 |
| Genetic reagent (*D. melanogaster*) | 12C07-GAL4 | Bloomington *Drosophila* Stock Center | RRID:BDSC_48496 | w[1118]; P{y[+t7.7] w[+mC]=GMR12C07-GAL4}attP2 |
| Genetic reagent (*D. melanogaster*) | 13B12-GAL4 | Bloomington *Drosophila* Stock Center | RRID:BDSC_45796 | w[1118]; P{y[+t7.7] w[+mC]=GMR13B12-GAL4}attP2 |
| Genetic reagent (*D. melanogaster*) | 15F10-GAL4 | Bloomington *Drosophila* Stock Center | RRID:BDSC_49266 | w[1118]; P{y[+t7.7] w[+mC]=GMR15F10-GAL4}attP2 |
| Genetic reagent (*D. melanogaster*) | 16C09-GAL4 | Bloomington *Drosophila* Stock Center | RRID:BDSC_48720 | w[1118]; P{y[+t7.7] w[+mC]=GMR16C09-GAL4}attP2 |
| Genetic reagent (*D. melanogaster*) | 21A01-GAL4 | Bloomington *Drosophila* Stock Center | RRID:BDSC_49853 | w[1118]; P{y[+t7.7] w[+mC]=GMR21A01-GAL4}attP2 |
| Genetic reagent (*D. melanogaster*) | 21C09-GAL4 | Bloomington *Drosophila* Stock Center | RRID:BDSC_48936 | w[1118]; P{y[+t7.7] w[+mC]=GMR21C09-GAL4}attP2 |
| Genetic reagent (*D. melanogaster*) | 24C04-GAL4 | Bloomington *Drosophila* Stock Center | RRID:BDSC_49072 | w[1118]; P{y[+t7.7] w[+mC]=GMR24C04-GAL4}attP2 |

*Continued on next page*

*Continued*

| Reagent type (species) or resource | Designation | Source or reference | Identifiers | Additional information |
|---|---|---|---|---|
| Genetic reagent (*D. melanogaster*) | 26B11-GAL4 | Bloomington *Drosophila* Stock Center | RRID:BDSC_49164 | w[1118]; P{y[+t7.7] w[+mC]=GMR26B11-GAL4}attP2 |
| Genetic reagent (*D. melanogaster*) | 26D04-GAL4 | Bloomington *Drosophila* Stock Center | RRID:BDSC_49175 | w[1118]; P{y[+t7.7] w[+mC]=GMR26D04-GAL4}attP2 |
| Genetic reagent (*D. melanogaster*) | 26F04-GAL4 | Bloomington *Drosophila* Stock Center | RRID:BDSC_49191 | w[1118]; P{y[+t7.7] w[+mC]=GMR26F04-GAL4}attP2 |
| Genetic reagent (*D. melanogaster*) | 35B08-GAL4 | Bloomington *Drosophila* Stock Center | RRID:BDSC_49818 | w[1118]; P{y[+t7.7] w[+mC]=GMR35B08-GAL4}attP2 |
| Genetic reagent (*D. melanogaster*) | 36C09-GAL4 | Bloomington *Drosophila* Stock Center | RRID:BDSC_49933 | w[1118]; P{y[+t7.7] w[+mC]=GMR36C09-GAL4}attP2 |
| Genetic reagent (*D. melanogaster*) | 37D11-GAL4 | Bloomington *Drosophila* Stock Center | RRID:BDSC_49536 | w[1118]; P{y[+t7.7] w[+mC]=GMR37D11-GAL4}attP2 |
| Genetic reagent (*D. melanogaster*) | 38H01-GAL4 | Bloomington *Drosophila* Stock Center | RRID:BDSC_50025 | w[1118]; P{y[+t7.7] w[+mC]=GMR38H01-GAL4}attP2 |
| Genetic reagent (*D. melanogaster*) | 39 F05-GAL4 | Bloomington *Drosophila* Stock Center | RRID:BDSC_50056 | w[1118]; P{y[+t7.7] w[+mC]=GMR39F05-GAL4}attP2 |
| Genetic reagent (*D. melanogaster*) | 42G08-GAL4 | Bloomington *Drosophila* Stock Center | RRID:BDSC_50166 | w[1118]; P{y[+t7.7] w[+mC]=GMR42G08-GAL4}attP2 |
| Genetic reagent (*D. melanogaster*) | 44G12-GAL4 | Bloomington *Drosophila* Stock Center | RRID:BDSC_47933 | w[1118]; P{y[+t7.7] w[+mC]=GMR44G12-GAL4}attP2 |
| Genetic reagent (*D. melanogaster*) | 44H11-GAL4 | Bloomington *Drosophila* Stock Center | RRID:BDSC_41268 | w[1118]; P{y[+t7.7] w[+mC]=GMR44H11-GAL4}attP2 |
| Genetic reagent (*D. melanogaster*) | 45D07-GAL4 | Bloomington *Drosophila* Stock Center | RRID:BDSC_49562 | w[1118]; P{y[+t7.7] w[+mC]=GMR45D07-GAL4}attP2 |
| Genetic reagent (*D. melanogaster*) | 48H11-GAL4 | Bloomington *Drosophila* Stock Center | RRID:BDSC_50396 | w[1118]; P{y[+t7.7] w[+mC]=GMR48H11-GAL4}attP2 |
| Genetic reagent (*D. melanogaster*) | 49F11-GAL4 | Bloomington *Drosophila* Stock Center | RRID:BDSC_38701 | w[1118]; P{y[+t7.7] w[+mC]=GMR49F11-GAL4}attP2 |
| Genetic reagent (*D. melanogaster*) | 54H12-GAL4 | Bloomington *Drosophila* Stock Center | RRID:BDSC_48205 | w[1118]; P{y[+t7.7] w[+mC]=GMR54 H12-GAL4}attP2/TM3, Sb[1] |
| Genetic reagent (*D. melanogaster*) | 57F03-GAL4 | Bloomington *Drosophila* Stock Center | RRID:BDSC_46386 | w[1118]; P{y[+t7.7] w[+mC]=GMR57F03-GAL4}attP2 |
| Genetic reagent (*D. melanogaster*) | 60B12-GAL4 | Bloomington *Drosophila* Stock Center | RRID:BDSC_39239 | w[1118]; P{y[+t7.7] w[+mC]=GMR60B12-GAL4}attP2 |

*Continued*

| Reagent type (species) or resource | Designation | Source or reference | Identifiers | Additional information |
|---|---|---|---|---|
| Genetic reagent (*D. melanogaster*) | 60D12-GAL4 | Bloomington *Drosophila* Stock Center | RRID:BDSC_39249 | w[1118]; P{y[+t7.7] w[+mC]=GMR60D12-GAL4}attP2 |
| Genetic reagent (*D. melanogaster*) | 60G04-GAL4 | Bloomington *Drosophila* Stock Center | RRID:BDSC_39258 | w[1118]; P{y[+t7.7] w[+mC]=GMR60G04-GAL4}attP2 |
| Genetic reagent (*D. melanogaster*) | 64C04-GAL4 | Bloomington *Drosophila* Stock Center | RRID:BDSC_39296 | w[1118]; P{y[+t7.7] w[+mC]=GMR64C04-GAL4}attP2 |
| Genetic reagent (*D. melanogaster*) | 70G12-GAL4 | Bloomington *Drosophila* Stock Center | RRID:BDSC_39552 | w[1118]; P{y[+t7.7] w[+mC]=GMR70G12-GAL4}attP2 |
| Genetic reagent (*D. melanogaster*) | 72C01-GAL4 | Bloomington *Drosophila* Stock Center | RRID:BDSC_47729 | w[1118]; P{y[+t7.7] w[+mC]=GMR72C01-GAL4}attP2 |
| Genetic reagent (*D. melanogaster*) | 73F02-GAL4 | Bloomington *Drosophila* Stock Center | RRID:BDSC_39824 | w[1118]; P{y[+t7.7] w[+mC]=GMR73F02-GAL4}attP2 |
| Genetic reagent (*D. melanogaster*) | 75B09-GAL4 | Bloomington *Drosophila* Stock Center | RRID:BDSC_39883 | w[1118]; P{y[+t7.7] w[+mC]=GMR75B09-GAL4}attP2 |
| Genetic reagent (*D. melanogaster*) | 76E12-GAL4 | Bloomington *Drosophila* Stock Center | RRID:BDSC_47753 | w[1118]; P{y[+t7.7] w[+mC]=GMR76E12-GAL4}attP2 |
| Genetic reagent (*D. melanogaster*) | 79G12-GAL4 | Bloomington *Drosophila* Stock Center | RRID:BDSC_40051 | w[1118]; P{y[+t7.7] w[+mC]=GMR79G12-GAL4}attP2 |
| Genetic reagent (*D. melanogaster*) | 83B04-GAL4 | Bloomington *Drosophila* Stock Center | RRID:BDSC_41309 | w[1118]; P{y[+t7.7] w[+mC]=GMR83B04-GAL4}attP2 |
| Genetic reagent (*D. melanogaster*) | nompC-GAL4 | Bloomington *Drosophila* Stock Center | RRID:BDSC_36361 | y[1] w[*]; PBac{y[+mDint2] w[+mC]=nompC GAL4.P}VK00014; Df(3 L)Ly, sens[Ly-1]/TM6C, Sb[1] Tb[1] |
| Genetic reagent (*D. melanogaster*) | UAS-mCD8::GFP | Gift from Rubin Lab, Janelia | Gift from Rubin Lab, Janelia | P{pJFRC7-020XUAS-IVSmCD8::GFP}attP2 |

## Resource availability

### Lead contact

Further information and requests for resources and reagents should be directed to and will be fulfilled by the lead contact, John C. Tuthill (tuthill@uw.edu).

### Materials availability

The genetic driver lines used in this study are listed in *Appendix 1—table 3* and are available from the Bloomington *Drosophila* Stock center.

## EM images & neuron reconstruction

The 3D reconstructed axons are from the FANC dataset (*Phelps et al., 2021*), for details on segmentation, see *Azevedo et al., 2024*. Only the left wing afferents were analyzed due to damage to the right side ADMN (Azevedo et al., Extended Data *Figure 4*). Following automatic segmentation, neurons were proofread to include primary neurites and as many branches as could confidently be reattached. Neurons were annotated using CAVE (*Dorkenwald et al., 2025*). Depth-colored reconstructions were created using braincircuits.io (*Azevedo et al., 2024*).

## Reconstructed axon morphology clusters

To group axons by similar connectivity, we computed the cosine similarity of synaptic weights onto postsynaptic partners. We included fragments (9.7% of total output synapses) and used a three-synapse threshold for connections. Cosine similarity and agglomerative clustering were computed with the Python library Scikit-learn (cosine_similarity, AgglomerativeClustering, and dendrogram packages) (*Pedregosa et al., 2011*). A permutation test to compare within- and between-cluster similarity was computed with the Python library SciPy (*Virtanen et al., 2020*). Information on synapse location predictions and error can be found in *Azevedo et al., 2024*.

## Animals

We used *Drosophila melanogaster* raised on standard cornmeal, molasses, and yeast medium at 25 °C in a 14:10 hr light:dark cycle. We used female flies 2–7 days post-eclosion for imaging. The genetic driver lines screened are listed in *Appendix 1—table 3*. Originally, they were chosen by manually looking through the FlyLight database for driver lines with sparse expression in the ADMN. Later, the braincircuits.io 'genetic lines matching' tool was used to screen driver lines by inputting segment IDs for particular wing afferent morphologies, with the particular lines selected based on sparse ADMN expression.

The fly food recipe used was based on the Bloomington standard Cornmeal, Molasses, and Yeast Medium recipe, which can be found at https://bdsc.indiana.edu/information/recipes/molassesfood.html. Our recipe had only slightly different antifungal ingredients and included tegosept, propionic acid, and phosphoric acid.

## Sample preparation

### Wing images

To remove wings, flies were briefly anesthetized using $CO_2$ before using forceps to delicately cut around the wing hinge and remove the wing with the sclerites that make up the wing hinge intact. One wing was collected from 4 to 6 females to ensure that expression was consistent across individuals. Wings were then fixed in 4% paraformaldehyde (PFA) PBS solution for 20–60 min. Next, wings were rinsed in PBS with 0.2% Triton X-100 (PBT) four times over the course of 75 min. For most samples, native fluorescence was imaged, so the wings were then mounted onto slides in Vectashield without DAPI.

For preparations requiring phalloidin staining to label cap cells of chordotonal organs, after rinsing, wings were incubated in 1:50 Alexa Fluor 647 nm Phalloidin (Thermo Fisher A22287) in a PBS solution with the following reagents to improve tissue penetrance: 1% triton X-100, 0.5% DMSO, 0.05 mg/ml Escin (Sigma-Aldrich, E1378), and 3% normal goat serum. Wings were then incubated for ten days at 4 °C overnight with gentle nutating at room temperature during the day. Following incubation, a second rinsing procedure was performed (four washes in PBT over the course of 75 min) before mounting the wings on slides with Vectashield, as above.

### Wing hinge images

For wing hinge images, a full adult fly was hemisected. First, flies were sacrificed by chilling briefly on ice, then dipping in 95% ethanol. Next, they were frozen in Tissue-Tek O.C.T. Compound on dry ice for ~3 min. Flies were then sliced along the anterior-posterior axis with a razor blade and transferred to a series of wells of ~3 mL 4% paraformaldehyde PBS solution until the O.C.T. melted away. Hemisected flies were then transferred to a 0.6 mL tube with fresh fixative for 45 min before following the same washing procedure detailed above. Instead of Vectashield, hemisected flies were mounted using the FocusClear-MountClear system (CelExplorer FC-101 and MC-301).

## Confocal imaging and image post-processing

Mounted wings and wing hinges were imaged on a Confocal Olympus FV1000. Images were processed in FIJI (*Schindelin et al., 2012*).

### FlyLight confocal stacks

Confocal stacks were downloaded from the gen1 GAL4 and MCFO GAL4 collections on FlyLight (*Jenett et al., 2012*; *Meissner et al., 2023*) and displayed as max projections using FIJI. All FlyLight

Z-stacks for the genotypes in this project are publicly available online at https://www.janelia.org/project-team/flylight. For *Figures 3, 5 and 6*, VNC expression patterns from the full GAL4 lines were aligned using the Computational Morphometry Toolkit (CMTK) to a female VNC template (*Bogovic et al., 2020*) in FIJI. For *Figures 4 and 7*, MCFO images were used because the full expression patterns were too broad in the whole VNC to visualize the wing sensory neurons in a max projection.

The depth-colored FlyLight MCFO images in *Figure 8* were adjusted to visually highlight single neurons. First, we duplicated the max-projection Z-stack and increased the contrast on one copy. Next, we traced the relevant neuron in the original and used this shape to mask the high-contrast copy. We then overlaid this masked image onto the original. This method allowed us to highlight single neurons visually in busy MCFO images. Full Z-stacks are available through FlyLight.

## Peripheral identification

See *Appendix 1—table 2* for a list of references we used to identify peripheral structures along the wing and near the wing hinge (*Cole and Palka, 1982*; *Dinges et al., 2021*; *Fudalewicz-Niemczyk, 1963*; *Hartenstein and Posakony, 1989*; *Hertweck, 1931*). Sensory structures were identified from confocal image stacks by closely scrutinizing the images to see exactly where GFP-labeled neurons were in relation to landmarks, such as wing veins and sclerites. Campaniform sensilla were the most straightforward sensory structures to identify thanks to a comprehensive atlas (*Dinges et al., 2021*). The chordotonal organs were identified by their actin-rich attachment cells labeled by phalloidin. The structure on the tegula was identified as a HP due to the appearance of the hairs.

## Acknowledgements

We thank Sweta Agrawal, Bradley Dickerson, Michael Dickinson, and members of the Dickinson and Tuthill Labs for comments on the manuscript and thoughtful discussions on all things wing. We especially thank Anne Sustar, Leila Elabbady, and Brandon Pratt for their valuable feedback on an early draft. Stocks obtained from the Bloomington *Drosophila* Stock Center (NIH P40OD018537) were used in this study. This project was supported by National Institutes of Health training grants T32NS099578 and T90DA032436 to EL, a Pecot Fellowship award from the McKnight Foundation to AM, as well as National Institutes of Health Grants U19NS104655 and R01NS102333, a Searle Scholar Award, a Klingenstein-Simons Fellowship, a Pew Biomedical Scholar Award, a McKnight Scholar Award, a Sloan Research Fellowship, and the New York Stem Cell Foundation to JCT. JCT is a New York Stem Cell Foundation – Robertson Investigator.

## Additional information

### Competing interests

John C Tuthill: Reviewing editor, eLife. The other authors declare that no competing interests exist.

### Funding

| Funder | Grant reference number | Author |
| --- | --- | --- |
| National Institute of Neurological Disorders and Stroke | U19NS104655 | Ellen Lesser<br>Anthony Moussa<br>John C Tuthill |
| National Institute of Neurological Disorders and Stroke | R01NS102333 | Ellen Lesser<br>Anthony Moussa<br>John C Tuthill |
| McKnight Foundation | Pecot Fellowship | Anthony J Moussa |
| National Institutes of Health | T90DA032436 | Ellen Lesser |
| National Institutes of Health | T32NS099578 | Ellen Lesser |

| Funder | Grant reference number | Author |
|---|---|---|
| New York Stem Cell Foundation | Robertson Neuroscience Investigator Award | Ellen Lesser<br>Anthony J Moussa<br>John C Tuthill |

The funders had no role in study design, data collection and interpretation, or the decision to submit the work for publication.

## Author contributions

Ellen Lesser, Conceptualization, Data curation, Formal analysis, Funding acquisition, Validation, Investigation, Visualization, Methodology, Writing – original draft, Writing – review and editing; Anthony J Moussa, Funding acquisition, Validation, Investigation, Visualization; John C Tuthill, Conceptualization, Funding acquisition, Writing – original draft, Project administration, Writing – review and editing

## Author ORCIDs

Ellen Lesser ⬢ https://orcid.org/0000-0001-7929-0503
Anthony J Moussa ⬢ https://orcid.org/0000-0002-9538-2747
John C Tuthill ⬢ https://orcid.org/0000-0002-5689-5806

Reviewer #1 (Public review): https://doi.org/10.7554/eLife.107867.3.sa1
Reviewer #2 (Public review): https://doi.org/10.7554/eLife.107867.3.sa2
Reviewer #3 (Public review): https://doi.org/10.7554/eLife.107867.3.sa3
Author response https://doi.org/10.7554/eLife.107867.3.sa4

## Additional files

### Supplementary files

MDAR checklist

### Data availability

VNC images are publicly available via FlyLight (https://www.janelia.org/project-team/flylight). Confocal stacks of the genetic expression in the wing for each driver line are available for download from Dryad (https://doi.org/10.5061/dryad.mgqnk99b5). An annotation table that includes the FANC cell ID and peripheral identification of each segment in detail is available to the FANC community, as well as on Dryad as a CSV. Analyses and a connectivity table are stored at https://github.com/EllenLesser/Lesser_eLife_2025 (copy archived at *Lesser, 2026*).

The following dataset was generated:

| Author(s) | Year | Dataset title | Dataset URL | Database and Identifier |
|---|---|---|---|---|
| Tuthill JC | 2025 | Peripheral anatomy and central connectivity of proprioceptive sensory neurons in the *Drosophila* wing | https://doi.org/10.5061/dryad.mgqnk99b5 | Dryad Digital Repository, 10.5061/dryad.mgqnk99b5 |

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

# Appendix 1

**Appendix 1—table 1.** ADMN sensory axon identification and nomenclature.

| Sensory structure | Axon morphology identification | MANC connectome nomenclature (Marin et al., 2024) |
|---|---|---|
| Small campaniform sensilla (proximal) | *Palka et al., 1979*; *Ghysen, 1980* | SApp04, SApp10, SApp11, SApp13, SApp14, SApp18, SApp19, SApp20, SApp21 |
| Small campaniform sensilla (distal) | *Palka et al., 1979*; *Ghysen, 1980* | SNpp04, SNpp08, SNpp11, SNpp33, SNpp36, SNpp06, SNpp26 |
| Large campaniform sensilla | *Burt and Palka, 1982*; *Palka et al., 1986* | SNpp30, SNpp32, SNpp31 |
| Tegula campaniform sensilla | *Lesser et al., 2024* | SNpp28, SNpp37, SNpp38 |
| Tegula hair plate | *Lesser et al., 2024* | SNxx26 |
| Tegula chordotonal organ | *Lesser et al., 2024* | SNpp07, SNpp10 |
| Radius chordotonal organ | *Lesser et al., 2024* | SNpp29, SNpp61, SNpp62, SNpp63 |
| Thorax sensor | *Lesser et al., 2024* | SNpp16 |
| Thorax macrochete | *Ghysen, 1980*; *Kays et al., 2014* | SNta05, SNta10, SNta12, SNta13 |
| Thorax microchete | *Ghysen, 1978*; *Ghysen, 1980* | SNta01, SNta02, SNta09 |
| Margin mechanosensors | *Palka et al., 1979* | SNta04, SNta06, SNta07, SNta08, SNta11, SNta14, SNta18 |
| Margin chemosensors | *Lu et al., 2012*; *Thistle et al., 2012*; *Koh et al., 2014* | SNch02, SNch03, SNch04, SNch12 |
| Unknown | - | SNpp05, SNpp09, SNpp13, SNpp27, SNxx28, SNtaxx, SNxxxx, SNxx24, SNxx25 |

**Appendix 1—table 2.** Literature characterizing peripheral anatomy of wing mechanosensory neurons.

| Paper | Structures identified |
|---|---|
| *Hertweck, 1931* | Chordotonal organ in the radius, thorax sensor. |
| *Fudalewicz-Niemczyk, 1963* | Neuronal innervation in wings of ten dipteran species. |
| *Cole and Palka, 1982* | Differences in peripheral morphologies of wing campaniform sensilla domes; Homologies between wing campaniform sensilla and haltere campaniform sensilla. |
| *Hartenstein and Posakony, 1989* | Comprehensive identification of wing and thorax bristles throughout development. |
| *Dinges et al., 2021* | Comprehensive atlas of all campaniform sensilla in *Drosophila melanogaster*. |

**Appendix 1—table 3.** Driver lines labeling wing sensory neurons.
Numbers in the table indicate how many neurons are labeled, e.g., 4 of 24 radius chordotonal organs (CO) neurons for the first driver line, 10A07-GAL4.

| Peripheral structure | total | Radius CO | Tegula CO | ANWP CS | Tegula CS field | d.Rad.A | d.Rad.B | d.Rad.C | v.Rad.A | v.Rad.B | v.Rad.C | d.Rad.D | d.Rad.E | dS-1 & 2 | d.HCV | GSR | TSM-1 & 2 | ACV | L3-1 | L3-2 | L3-3 | v.HCV | L3-V | Thorax receptor | Tegula hair plate | Tegula hairs |
|---|---|---|---|---|---|---|---|---|---|---|---|---|---|---|---|---|---|---|---|---|---|---|---|---|---|---|
| # of neurons | total | ~24 | ~14 | 3 | 18 | 4 | 7 | 18 | 4 | 3 | 5 | 4 | 8 | 2 | 1 | 1 | 2 | 1 | 1 | 1 | 1 | 1 | 1 | 5 | 5 | 6 |
| 10A07-Gal4 | 4 | 4 | | | | | | | | | | | | | | | | | | | | | | | | |
| 10F07-Gal4 | 8 | 7 | | | | 1 | | | | | | | | | | | | | | | | | | | | |
| 10G03-Gal4 | 9 | | | | | | | 2 | 2 | | | | | | | | | | | | | | | 5 | | |
| 12C07-Gal4 | 19 | | | | | | 2 | 10 | 1 | 3 | | 2 | | | | | | 1 | | | | | | | | |
| 13B12-Gal4 | 10 | | | | 8 | | | | | | | | | | | | | | | | | | | | 2 | |
| 15F10-Gal4 | 24 | | 24 | | | | | | | | | | | | | | | | | | | | | | | |
| 16C09-Gal4 | 4 | | | | | | | | | | | | | | | | | | | | | | | | 4 | |
| 21A01-Gal4 | 5 | | 5 | | | | | | | | | | | | | | | | | | | | | | | |
| 21C09-Gal4 | 7 | | | | | | | 2 | 4 | | 1 | | | | | | | | | | | | | | | |
| 24C04-Gal4 | 8 | 6 | | | | | | | | | 2 | | | | | | | | | | | | | | | |
| 26B11-Gal4 | 2 | | | | | | | | | | | | | | | | 2 | | | | | | | | | |
| 26D04-Gal4 | 12 | | | | | | | 5 | 2 | | 3 | | 2 | | | | | | | | | | | | | |
| 26F04-Gal4 | 36 | | | | | 4 | 7 | 17 | 2 | | 4 | | 2 | | | | | | | | | | | | | |
| 35B08-Gal4 | 3 | 3 | | | | | | | | | | | | | | | | | | | | | | | | |
| 36C09-Gal4 | 4 | 4 | | | | | | | | | | | | | | | | | | | | | | | | |
| 37D11-Gal4 | 8 | | | 1 | | | | | | | | | | | | | | | | | | | | 5 | | 2 |
| 38H01-Gal4 | 2 | | | | | | | | | | 1 | | | | | | | | | | | 1 | | | | |
| 39F05-Gal4 | 1 | 1 | | | | | | | | | | | | | | | | | | | | | | | | |
| 42G08-Gal4 | 2 | | | | 2 | | | | | | | | | | | | | | | | | | | | | |
| 44H11-Gal4 | 9 | 8 | 1 | | | | | | | | | | | | | | | | | | | | | | | |
| 48H11-Gal4 | 4 | 4 | | | | | | | | | | | | | | | | | | | | | | | | |
| 49F11-Gal4 | 7 | 7 | | | | | | | | | | | | | | | | | | | | | | | | |
| 54H12-Gal4 | 9 | | | 2 | 2 | | | | | | | | | 2 | | | | | | | | | | 3 | | |
| 57F03-Gal4 | 5 | | 5 | | | | | | | | | | | | | | | | | | | | | | | |
| 60B12-Gal4 | 4 | | | | | | | | | | | 2 | | | | | | | | 1 | 1 | | | | | |
| 60D12-Gal4 | 10 | | 10 | | | | | | | | | | | | | | | | | | | | | | | |
| 60G04-Gal4 | 5 | | 5 | | | | | | | | | | | | | | | | | | | | | | | |
| 64C04-Gal4 | 6 | 4 | | | | 1 | | 1 | | | | | | | | | | | | | | | | | | |
| 70G12-Gal4 | 2 | | | | | | | | | | | | | | | | | | | | | | | | 2 | |
| 72C01-Gal4 | 6 | 6 | | | | | | | | | | | | | | | | | | | | | | | | |
| 73F02-Gal4 | 2 | | | | 2 | | | | | | | | | | | | | | | | | | | | | |

*Appendix 1—table 3 continued*

| Peripheral structure | Radius CO | Tegula CO | ANWP CS | Tegula CS field | d.Rad.A | d.Rad.B | d.Rad.C | v.Rad.A | v.Rad.B | v.Rad.C | d.Rad.D | d.Rad.E | dS-1 & 2 | d.HCV | GSR | TSM-1 & 2 | ACV | L3-1 | L3-2 | L3-3 | v.HCV | L3-V | Thorax receptor | Tegula hair plate | Tegula hairs |
|---|---|---|---|---|---|---|---|---|---|---|---|---|---|---|---|---|---|---|---|---|---|---|---|---|---|
| 75B09-Gal4 | 21 | 4 | 2 | 13 | 2 | | | | | | | | | | | | | | | | | | | | |
| 76E12-Gal4 | 8 | 8 | | | | | | | | | | | | | | | | | | | | | | | |
| 79G12-Gal4 | 1 | | | | | | | | | 1 | | | | | | | | | | | | | | | |
| 83B04-Gal4 | 5 | | | | | | | | | | | | | | | | | | | | | | 5 | | |

