## [Editor Report · eLife Assessment]

This **important** work describes wing mechanosensory neurons in detail, extending our understanding of sensorimotor processing in the fruit fly. The evidence presented **convincingly** supports the authors' identification of these neurons and leverages state-of-the-art methods to generate a near-complete map of wing mechanosensory circuitry. Overall, this study provides new hypotheses and invaluable tools for investigating proprioceptive motor control of the wing in Drosophila.

---

## [Referee Report · Reviewer #1 (Public review)]

Summary:

Lesser et al provide a comprehensive description of Drosophila wing proprioceptive sensory neurons at the electron microscopy resolution. This "tour-de-force", provides a strong foundation for future structural and functional research aimed at understanding wing motor control in Drosophila with implications to understanding wing control across other insects.

Strengths:

(1) Authors leverage previous research that described many of the fly wing proprioceptors, and combine this knowledge with EM connectome data such that they now provide a near-complete morphological description of all wing proprioceptors.

(2) Authors cleverly leverage genetic tools and EM connectome data to tie the location of proprioceptors on the wings with axonal projections in the connectome. This enables them to both align with previous literature as well as make some novel claims.

(3) In addition to providing a full description of wing proprioceptors, authors also identified a novel population of sensors on the wing tegula that make direct connections with the B1 wing motor neurons implicating the role of tegula in wing movements that was previously underappreciated.

(4) Despite being the most comprehensive description so far, it is reassuring that authors clearly state the missing elements in the discussion.

Weaknesses:

(1) Authors do their main analysis on data from FANC connectome but provide corresponding IDs for sensory neurons in the MANC connectome. I wonder how the connectivity matrix compares across FANC and MANC if the authors perform similar analysis as they have done in Fig. 2. This could be a valuable addition and potentially also pick up any sexual dimorphism.

(2) Authors speculate about presence of gap junctions based on density of mitochondria. I'm not convinced about this given mitochondrial densities could reflect other things that correlate with energy demands in sub-compartments.

Overall, I consider this an exceptional analysis which will be extremely valuable to the community.

---

## [Referee Report · Reviewer #2 (Public review)]

Summary:

Lesser et al. present an atlas of Drosophila wing sensory neurons. They proofread the axons of all sensory neurons in the wing nerve of an existing electron microscopy dataset, the female adult fly nerve cord (FANC) connectome. These reconstructed sensory axons were linked with light microscopy images of full-scale morphology to identify their origin in the periphery of the wing and encoded sensory modalities. The authors described the morphology and postsynaptic targets of proprioceptive neurons as well as previously unknown sensory neurons.

Strengths:

The authors present a valuable catalogue of wing sensory neurons, including previously undescribed sensory axons in the Drosophila wing. By providing both connectivity information with linked genetic drive lines, this research facilitates future work on the wing motor-sensory network and applications relating to Drosophila flight. The findings were linked to previous research as well as their putative role in the proprioceptive and nerve cord circuitry, providing testable hypotheses for future studies.

Weaknesses:

With future use as an atlas, it should be noted that the evidence is based on sensory neurons on only one side of the nerve cord. Fruit flies have stereotyped left/right hemispheres in the brain and left/right hemisegments in the nerve cord. Comparison of left and right neurons of the nervous system can give a sense of how robust the morphological and connectivity findings are. Unfortunately, this dataset has damage to the right side, making such comparisons unreliable.

---

## [Referee Report · Reviewer #3 (Public review)]

Summary:

The authors aim to identify the peripheral end organ origin in the fly's wing of all sensory neurons in the Anterior Dorsal Mesothoracic nerve. They reconstruct the neurons and their downstream partners in an electron microscopy volume of a female ventral nerve cord, analyse the resulting connectome and identify their origin with review of the literature and imaging of genetic driver lines. While some of the neurons were already known through previous work, the authors expand on the identification and create a near complete map of the wing mechanosensory neurons at synapse resolution.

Strengths:

The authors elegantly combine electron microscopy neuron morphology, connectomics and light microscopy methods to bridge the gap between fly wing sensory neuron anatomy and ventral nerve cord morphology. Further, they use EM ultrastructural observations to make predictions on the signaling modality of some of the sensory neurons and thus their function in flight.

The work is as comprehensive as state of the art methods allow to create a near complete map of the wing mechanosensory neurons. This work will be of importance to the field of fly connectomics and modelling of fly behavior as well as a useful resource to the Drosophila research community.

Through this comprehensive mapping of neurons to the connectome the authors create a lot of hypotheses on neuronal function partially already confirmed with the literature and partially to be tested in the future. The authors achieved their aim of mapping the periphery of the fly's wing to axonal projections in the ventral nerve cord, beautifully laying out their results to support their mapping.

The authors identify the neurons in a previously published connectome of a male fly ventral nerve cord to enable cross-individual analysis of connections and find no indication of sexual dimorphism at the sensory neuron level. Further, together with their companion paper Dhawan et al., 2025 describing the haltere sensory neurons in the same EM dataset, they cover the entire mechanosensory space involved in Drosophila flight.

---

## [Author Response]

The following is the authors’ response to the original reviews.

**Public Reviews:**

**Reviewer #1 (Public review):**
Summary:Lesser et al provide a comprehensive description of Drosophila wing proprioceptive sensory neurons at the electron microscopy resolution. This “tour-de-force” provides a strong foundation for future structural and functional research aimed at understanding wing motor control in Drosophila with implications for understanding wing control across other insects.Strengths:(1) The authors leverage previous research that described many of the fly wing proprioceptors, and combine this knowledge with EM connectome data such that they now provide a near-complete morphological description of all wing proprioceptors.(2) The authors cleverly leverage genetic tools and EM connectome data to tie the location of proprioceptors on the wings with axonal projections in the connectome. This enables them to both align with previous literature as well as make some novel claims.(3) In addition to providing a full description of wing proprioceptors, the authors also identified a novel population of sensors on the wing tegula that make direct connections with the B1 wing motor neurons, implicating the role of the tegula in wing movements that was previously underappreciated.(4) Despite being the most comprehensive description so far, it is reassuring that the authors clearly state the missing elements in the discussion.Weaknesses:(1) The authors do their main analysis on data from the FANC connectome but provide corresponding IDs for sensory neurons in the MANC connectome. I wonder how the connectivity matrix compares across FANC and MANC if the authors perform a similar analysis to the one they have done in Figure 2. This could be a valuable addition and potentially also pick up any sexual dimorphism.

We agree that systematic comparisons will provide valuable insights as more connectome datasets become available. However, the primary goal of this study was to link central axon morphology with peripheral structures in the wing. We deliberately omitted more detailed and quantitative analyses of the downstream VNC circuitry, apart from providing a global view of the connectivity matrix and using it to cluster the sensory axon types. A more detailed and systematic comparison of wing sensorimotor circuit connectivity across different connectome datasets (FANC, MANC, BANC, IMAC) is the subject of ongoing work in our lab, which we feel is beyond the scope of this study. Here, we chose to match the wing proprioceptors to axons in MANC to demonstrate their stereotypy across individuals and to make them more accessible to other researchers. We found no obvious sexual dimorphism at the level of wing sensory neurons. We now note this in the Discussion.

(2) The authors speculate about the presence of gap junctions based on the density of mitochondria. I’m not convinced about this, given that mitochondrial densities could reflect other things that correlate with energy demands in sub-compartments.

We have moved speculation about mitochondria and gap junctions to the Discussion.

(3) I’m intrigued by how the tegula CO is negative for iav. I wonder if authors tried other CO labeling genes like nompc. And what does this mean for the nature of this CO. Some more discussion on this anomaly would be helpful.

Based on this suggestion, we have added an image showing that tegula CO neurons are labeled by nompC-Gal4.

(4) The authors conclude there are no proprioceptive neurons in sclerite pterale C based on Chat-Gal4 expression analysis. It would be much more rigorous if authors also tried a pan-neuronal driver like nsyb/elav or other neurotransmitter drivers (Vglut, GAD, etc) to really rule this out. (I hope I didn’t miss this somewhere.)

To address this, we imaged OK371-GFP, which labels glutamatergic neurons, in the wing and wing hinge. We saw expression in the wing, as others have reported (Neukomm et. al., 2014), but we saw no expression at the wing hinge. Apart from a handful of glutamatergic gustatory neurons in the leg, we are not aware of any other sensory neurons in the fly that are not labeled by Chat-Gal4.

Overall, I consider this an exceptional analysis that will be extremely valuable to the community.

We sincerely appreciate the reviewer’s positive feedback.

**Reviewer #2 (Public review):**
Summary:Lesser et al. present an atlas of Drosophila wing sensory neurons. They proofread the axons of all sensory neurons in the wing nerve of an existing electron microscopy dataset, the female adult fly nerve cord (FANC) connectome. These reconstructed sensory axons were linked with light microscopy images of full-scale morphology to identify their origin in the periphery of the wing and encoded sensory modalities. The authors described the morphology and postsynaptic targets of proprioceptive neurons as well as previously unknown sensory neurons.Strengths:The authors present a valuable catalogue of wing sensory neurons, including previously undescribed sensory axons in the Drosophila wing. By providing both connectivity information with linked genetic drive lines, this research facilitates future work on the wing motor-sensory network and applications relating to Drosophila flight. The findings were linked to previous research as well as their putative role in the proprioceptive and nerve cord circuitry, providing testable hypotheses for future studies.Weaknesses:(1) With future use as an atlas, it should be noted that the evidence is based on sensory neurons on only one side of the nerve cord. Fruit flies have stereotyped left/right hemispheres in the brain and left/right hemisegments in the nerve cord. The comparison of left and right neurons of the nervous system can give a sense of how robust the morphological and connectivity findings are. Here, the authors have not compared the left and right side sensory axons from the wing nerve, leaving potential for developmental variability across samples and left/right hemisegments.

The right ADMN nerve in the FANC dataset is partially severed, making left/right comparisons unreliable (see Azevedo 2024, Extended Data Figure 4). We have updated the text to explain this within the Methods section of the paper.

(2) Not all links between the EM reconstructions and driver lines are convincing. To strengthen these, for all EM-LM matches in Figures 3-7, rotated views of the driver line (matching the rotated EM views) should be shown to provide a clearer comparison of the data. In particular, Figure 3G and Figure 7B are not very convincing based on the images shown. MCFO imaging of the driver lines in Figure 3G and 7B would make this position stronger if a clone that matches the EM reconstruction could be identified.

Many of the Z-stack images in the paper are from the Janelia FlyLight collection, and unfortunately their imaging parameters were not optimized for orthogonal views. Rotated views are blurry and not especially helpful for comparison to EM reconstruction. We now point out in the text that interested readers can access the Z-stacks from FlyLight to see the dorsal-ventral projections.

Regarding Figure 3G and 7B, we have added markers to the image with corresponding descriptions in the legend to guide the reader through the image of the busy driver line. Although these lines label many cells in the VNC as a whole, they sparsely label cells in the ADMN, making them nonetheless useful for identifying peripheral sensory neurons.

(3) Figure 7B looks like the driver line might have stochastic expression in the sensory neuron, which further reduces confidence in the result shown in Figure 7C. Is this expression pattern in the wing consistently seen? Many split-GAL4s have stochastic expressions. The evidence would be strengthened if the authors presented multiple examples (~4-5) of each driver line’s expression pattern in the supplement.

Figure 7B shows sparse labeling of the driver line using the MCFO technique, as specified in the legend. Its unilateral expression is therefore not due to stochastic expression of the Gal4 line. We have added the “MFCO” label to the image to clarify.

(4) Certain claims in this work lack quantitative evidence. On line 128, for instance, “Overall, our comprehensive reconstruction revealed many morphological subgroups with overlapping postsynaptic partners, suggesting a high degree of integration within wing sensorimotor circuits.” If a claim of subgroups having shared postsynaptic partners is being made, there should have been quantitative evidence. For example, cosine similar amongst members of each group compared to the cosine similarity of shuffled/randomised sets of axons from different groups. The heat map of cosine similarity in Figure 2B alone is not sufficient.

We agree that illustrating the extent of shared postsynaptic partners across subgroups strengthens this point. We added a visualization showing pairwise similarity scores for within- and between-cluster neuron pairs (Figure 2B inset). We also performed a permutation test to determine that within-cluster similarity is significantly higher than between clusters, and we report the test in the results as well as the figure legend. This analysis provides a more quantitative summary of the qualitative trends in connectivity that are summarized in Figure 2B.

(5) Similarly, claims about putative electrical connections to b1 motor neurons are very speculative. The authors state that “their terminals contain very densely packed mitochondria compared to other cells”, without providing a quantitative comparison to other sensory axons. There is also no quantitative comparison to the one example of another putative electrical connection from the literature. Further, it should be noted that this connection from Trimarchi and Murphey, 1997, is also stated as putative on line 167, which further weakens this evidence. Quantification would strongly strengthen this position. Identification of an example of high mitochondrial density at a confirmed electrical connection would be even better. In the related discussion section “A potential metabolic specialization for flight circuitry”, it should be more clearly noted that the dense mitochondria could be unrelated to a putative electrical connection. If the authors have an alternative hypothesis about the mitochondria density, this should be stated as well.

We agree with the reviewer that the link between mitochondrial density and metabolic specialization is purely speculative in this context. Based on reviewer feedback, we have moved all mention of the relationship between mitochondrial density and gap junction coupling to the Discussion. We acknowledge that this may seem like a somewhat random and not quantitatively supported observation. However, we found the coincidence striking and worthy of mention, though it is only tangentially relevant to the rest of the paper. From conversations with colleagues, we have also heard that this relationship is consistent with as yet unpublished work in other model organisms (e.g., zebrafish, mouse).

The electrical coupling to b1 motor neurons is well-established (Fayyazuddin and Dickinson, 1999), and we have updated the text to state this more clearly. However, we agree that whether the specific neurons we have identified based on their anatomy are the same ones functionally identified through whole-nerve recordings remains unknown.

(6) It would be appropriate to cite previous work using a similar strategy to match sensory axons to their cell bodies/dendrites at the periphery using driver lines and connectomics (see Figure 5 for example in the following paper: https://doi.org/10.7554/eLife.40247).

At this point, there are now dozens of papers that match the axons of sensory neurons to their cell bodies/dendrites in the periphery by comparing light microscopy and connectomics. When we dug in, we found examples in *C. elegans*, *Ciona intestinalis*, zebrafish, and mouse, all published prior to the study cited above. For basically every animal for which scientists have acquired EM volumes of neural tissue, they have used other anatomical labeling methods to determine cell types inside and outside the imaged volume. In summary, we found it difficult to establish a single primary citation for this approach. In lieu of this, we have added a citation to an earlier review by a pioneer in EM connectomics that discusses the general approach of matching cells across different labeling/imaging modalities (Meinertzhagen et al., 2009).

The methods section is very sparse. For the sake of replicability, all sections should be expanded upon.

We have expanded the methods section, and also a STAR methods table.

**Reviewer #3 (Public review):**
Summary:The authors aim to identify the peripheral end-organ origin in the fly’s wing of all sensory neurons in the anterior dorsomedial nerve. They reconstruct the neurons and their downstream partners in an electron microscopy volume of a female ventral nerve cord, analyse the resulting connectome, and identify their origin with a review of the literature and imaging of genetic driver lines. While some of the neurons were already known through previous work, the authors expand on the identification and create a near-complete map of the wing mechanosensory neurons at synapse resolution.Strengths:The authors elegantly combine electron microscopy, neuron morphology, connectomics, and light microscopy methods to bridge the gap between fly wing sensory neuron anatomy and ventral nerve cord morphology. Further, they use EM ultrastructural observations to make predictions on the signaling modality of some of the sensory neurons and thus their function in flight.The work is as comprehensive as state-of-the-art methods allow to create a near-complete mapof the wing mechanosensory neurons. This work will be of importance to the field of fly connectomics and modelling of fly behavior, as well as a useful resource to the Drosophila research community.Through this comprehensive mapping of neurons to the connectome, the authors create a lot of hypotheses on neuronal function, partially already confirmed with the literature and partially to be tested in the future. The authors achieved their aim of mapping the periphery of the fly’s wing to axonal projections in the ventral nerve cord, beautifully laying out their results to support their mapping.The authors identify the neurons in a previously published connectome of a male fly ventral nerve cord to enable cross-individual analysis of connections. Further, together with their companion paper, Dhawan et al. 2025, describing the haltere sensory neurons in the same EM dataset, they cover the entire mechanosensory space involved in Drosophila flight.Weaknesses:The connectomic data are only available upon request; the inclusion of a connectivity table of the reconstructed neurons would aid analysis reproducibility and cross-dataset comparisons.

We have added a connectivity table as well as analysis scripts in the github repository for the paper (https://github.com/EllenLesser/Lesser_eLife_2025).

**Recommendations for the authors:**

**Reviewer #2 (Recommendations for the authors):**
The methods section should be expanded in every aspect. Most pressing sections are:(1) Data and Code availability: All code should be included as a Zenodo database, the suggestion to ask authors for code upon request is inappropriate.

We have added all code to a public github repository, which is now linked in the Methods section.

(2) Samples: Standard cornmeal and molasses medium should have a reference, as many institutes use different recipes.

The recipe used by the University of Washington fly kitchen is based on the Bloomington standard Cornmeal, Molasses and Yeast Medium recipe, which can be found at https://bdsc.indiana.edu/information/recipes/molassesfood.html. The UW recipe is slightly modified for different antifungal ingredients and includes tegosept, propionic acid, and phosophoric acid.

(3) Table 3: Driver lines labelling wing sensory neurons: The genetic driver lines should have associated Bloomington stock centre numbers. Additionally, relevant information for effector lines used should be included in the methods.

We now include the Bloomington stock numbers and more information on effector lines in the STAR methods table.

Minor corrections:(1) Lines 119-120: “Notably, many of the axons do not form crisp cluster boundaries, suggesting that multimodal sensory information is integrated at early stages of sensory processing.” We do not follow the logic of this statement and suspect it is a bit too speculative.

We removed this sentence from the manuscript.

(2) Figure 1: The ADMN is missing in the schematics and would be helpful to depict for non-experts. Is this what is highlighted in Figure 1D?

Yes, and we now label 1D as the ADMN wing nerve.

(3) Figure 1B: Which driver lines are being depicted here? Looking at Table 3 does not clarify. It should be specified at least in the figure legend.

As stated in the legend, we include a table of all of the driver lines we screened and which sensory structures they label.

(4) Figure 1C: There are some minor placement issues with the text in the schematic. There is an arrow very close to the “CO” on the top right, which makes the “O” look like the symbol for male. “ax ii” is a bit too close to the wing hinge

We updated the figure to address this issue.

(5) Figure 1D: The outlined grey masks are not clear. The use of colour would be very useful for the reader to help understand what the authors are referring to here

We now use color for the masks.

(6) Figure 2A: It is unclear if the descending neuron and non-motor efferent neuron are not shown because they are under the described threshold, or to simplify the plot. They should be included in the plot if over the threshold.

We have updated the legend to specify that the exclusion of the descending and non-motor efferent neurons are to visually simplify the plot. We include % of sensory output to each of these neurons in the legend, and they are included in the connectivity matrix data in the public GitHub repository associated with the paper, included in the Methods.

(7) Figure 2B: What clustering is used specifically? The method says it’s from Scikit-learn, but there are many types of clustering available in this package.

We now include the specific clustering type used in the Methods section, which is agglomerative clustering.

(8) Figure 3A: What does the green box behind the plot represent?

The green box represents the tegula CO axons, which we now specify in the legend.

(9) Figure 3C: the “C” is clipped at the top.

We updated the figure to address this issue.

(10) Figure 4A: the main text says a “group of four axons” (line 203) while the figure says 5 axons.

We updated the text to address this issue.

(11) Line 360: “We found that the campaniform sensilla on the tegula provide the most direct feedback onto wing steering motor neurons”. We struggled to find where this was directly shown, because several sensory axon types directly synapse onto motor neurons.

We now specify in the text that this finding is shown in Figure 3.

**Reviewer #3 (Recommendations for the authors):**
I would like to congratulate the authors on their beautiful, easy-to-read, and easy-to-comprehend manuscript, with clear figures and nice visualizations. This work provides a valuable resource that will contribute to the interpretability of connectomic data and further to connectome-based modeling of fly behavior.

We sincerely appreciate the reviewer’s positive feedback.